# Holocene atmospheric iodine evolution over the North Atlantic

Juan Pablo Corella[1], Niccolo Maffezzoli[2,3], Carlos Alberto Cuevas[1], Paul Vallelonga[2], Andrea Spolaor[3], Giulio Cozzi[3], Juliane Müller[4,5], Bo Vinther[2], Carlo Barbante[3,6], Helle Astrid Kjær[2], Ross Edwards[7,8] and Alfonso Saiz-Lopez[1*]

[1]Department of Atmospheric Chemistry and Climate, Institute of Physical Chemistry Rocasolano, CSIC, Serrano 119, 28006 Madrid, Spain

[2]Centre for Ice and Climate, Niels Bohr Institute, University of Copenhagen, Juliane Maries vej 30, Copenhagen Ø 2100, Denmark

[3]Institute of Polar Sciences, ISP-CNR, Via Torino 155, 30170 Venice, Italy

[4]Alfred Wegener Institute, Helmholtz Center for Polar and Marine Research, Am Alten Hafen 26, 27568 Bremerhaven, Germany

[5]MARUM Research Faculty, University of Bremen, Leobener Strasse 8, 28359 Bremen, Germany

[6]Ca´Foscari University of Venice, Department of Environmental Sciences, Informatics and Statistics, Via Torino 155, 30170 Venice, Italy

[7]Physics and Astronomy, Curtin University, Kent St, Bentley WA 6102, Australia

[8]Department of Civil and Environmental Engineering, UW-Madison, Madison, WI 53706, USA

*Correspondence to*: A. Saiz-Lopez (a.saiz@csic.es)

**Abstract.** Atmospheric iodine chemistry has a large influence on the oxidizing capacity and associated radiative impacts in the troposphere. However, information on the evolution of past atmospheric iodine levels is restricted to the Industrial Period while its long-term natural variability remains unknown. The current levels of iodine in the atmosphere are controlled by anthropogenic ozone deposition to the ocean surface. Here, using high-resolution geochemical measurements from coastal eastern Greenland ReCAP (REnland ice CAP project) ice core, we report the first record of atmospheric iodine variability in the North Atlantic during the Holocene (i.e. the last 11,700 years). Surprisingly, our results reveal that the highest iodine concentrations in the record were found during the Holocene Thermal Maximum (HTM; ~11,500-5,500 years before -present). These high iodine levels could be driven by marine primary productivity resulting in an Early Holocene "Biological Iodine Explosion". The high and stable iodine levels during this past warm period is a useful observational constraint on projections of future changes in Arctic atmospheric composition and climate resulting from global warming.

## 1 Introduction

The integration of environmental proxies from different natural archives allows for the detailed understanding of climate variability during the Holocene (last 11,7 kyr BP) (*Mayewski et al.*, 2004). Nevertheless, there is surprisingly little systematic knowledge about atmospheric chemistry during this period, which is a key factor for understanding the background of natural

variability underlying anthropogenic climate change (IPCC, 2013). Global atmospheric models have recently shown the significant contribution of halogen chemistry to the oxidizing capacity of the atmosphere and associated radiative impacts (*Hossaini et al.*, 2015; *Saiz-Lopez et al.*, 2012a, 2014; *Sherwen et al.*, 2016, 2017). Reactive halogens containing chlorine,

bromine and iodine atoms cause ozone depletion through efficient catalytic cycles (*Saiz-Lopez and von Glasow*, 2012, *Simpson et al., 2015*). In particular, atmospheric reactive iodine is responsible for up to 27% of the total ozone loss in the marine boundary layer and upper troposphere (*Saiz-Lopez et al.*, 2014). Iodine-mediated ozone depletion negatively contributes to the longwave radiative flux in the troposphere (*Hossaini et al.*, 2015; *Saiz-Lopez et al.*, 2012a; *Sherwen et al., 2017*). Reactive iodine could also be involved in atmospheric particle formation (*Allan et al.*, 2015; *Roscoe et al.*, 2015; S*ipilä et al.*, 2016),

which has been suggested to have potential climatic implications in the troposphere.

The origins and cycling of iodine in the global atmosphere involve ocean emissions from inorganic (hypoiodous acid, HOI and molecular iodine, $I_2$) and very short lived (VSL) organic sources ($CH_3I$, $CH_2I_2$, $CH_2ICl$ and $CH_2IBr$)). Globally, the main source of present-day atmospheric iodine is the inorganic emission of HOI and $I_2$ from the ocean surface as a product of the reaction of iodide with deposited ozone (*Carpenter et al.*, 2013; *MacDonald et al.*, 2014; *Prados-Roman et al.*, 2015).

Biota in the marine environment are known to produce alkyl iodides, which are a primary source of reactive iodine due to their short photolysis lifetimes (from minutes to several days) (*Saiz-Lopez et al.*, 2012b). In polar regions, another important source of reactive iodine is the biogenic production of HOI and $I^-$ from algae underneath the sea ice (in equilibrium with $I_2$ and $H_2O$), and its subsequent diffusion through brine channels to the overlying atmosphere (*Saiz-Lopez et al.*, 2015). Other proposed mechanisms for iodine reaction in ice include the production of $I_2$ and tri-iodide ($I_3^-$) through the photo-oxidation of iodide

(*Kim et al.*, 2016), and the heterogeneous photo-reduction of iodate (*Gálvez et al.*, 2016). The first measurements of $I_2$ in the Arctic atmosphere and iodide in the Arctic snowpack have been recently reported by Raso et al., (2017). The recycling of iodine on ice/snow surfaces represents an offset change in partitioning ultimately increasing the effective atmospheric lifetime of iodine against deposition (*Saiz-Lopez et al.*, 2014). Particle-bound iodine compounds related to terrestrial biogenic material and mineral dust might also contribute to the total atmospheric iodine concentrations (*Spolaor et al.*, 2013). Sea spray aerosols

(ssa) expelled from the ocean surface during wave breaking also incorporate iodine in their structure, which can be subsequently released to the gas phase mainly due to the photolysis of the diatomic ICl and IBr species recycled by heterogeneous reactions over ssa (McFiggans et al., 2000; *Saiz-Lopez et al.*, 2014).

The polar ice sheets continuously archive atmospheric composition, and ice core records can be used to reconstruct atmospheric iodine deposition at centennial to millennial time-scales. Ice core iodine reconstructions in polar regions are

limited to three coastal sites: Law Dome (Antarctica) (*Vallelonga et al.*, 2017), Talos Dome (Antarctica) *(Spolaor et al., 2013)* and Severnaya Zemlya (Russian Arctic) (*Spoloar et al.*, 2016). Iodine variability recorded at these sites is strongly affected by regional sea ice dynamics since sea ice bioproductivity is thought to be one of the main sources for Antarctic and Arctic atmospheric iodine (*Saiz-Lopez et al.*, 2015). Unfortunately, the iodine records extracted from Law Dome and Severnaya Zemlya barely span the past few decades, thus preventing the assessment of millennial-scale atmospheric iodine variability

and its source emission mechanisms. The only iodine record beyond the Industrial Period available to date is restricted to the

Talos Dome site, approximately 250 km away from the Ross Sea and the Indian Ocean (East Antarctica) (*Spolaor et al.*, 2013), which provides a reconstruction of iodine variability for the last two glacial cycles. Sea ice dynamics, algal productivity and dust variability in the Antarctic region controlled iodine variability at glacial-interglacial timescales (*Spolaor et al.*, 2013). Unfortunately, due to the low temporal resolution (mean resolution >2 kyrs/sample) and the lack of iodine data during the last four millennia, this record does not allow for a reconstruction of iodine variability and related environmental drivers during the Holocene.

The ReCAP ice core, drilled from the coastal Renland ice cap, in East Greenland (Fig. 1) constitutes the only Greenland ice core with a complete Holocene climate stratigraphy largely undisturbed by glaciological "brittle ice" effects (*Vinther et al.*, 2009). The Renland ice cap is an ideal location for investigating ocean trace gas emissions since air masses feeding this region are mostly sourced from the North Atlantic Ocean, from 50ºN up to the Fram Strait (*Maffezzoli et al.*, 2018). From the analysis of the upper 130 m of the ReCAP ice core (CE 1750-2011), Cuevas et al (2018) have shown that a threefold increase in the North Atlantic iodine concentration has occurred during the last six decades. This increase is mainly driven by anthropogenic ozone pollution and enhanced sub-ice phytoplankton production associated with the recent thinning of Arctic sea ice (*Cuevas et al.*, 2018). This threefold increase has also been recorded in an alpine ice core (Col du Dome ice core), which likely records iodine emissions from the Mediterranean Sea (*Legrand et al.*, 2018). The recent increase in anthropogenic ozone in the troposphere and its subsequent deposition onto the ocean surface favours the tropospheric ozone reaction with iodide ions, accelerating the release of HOI and $I_2$ to the overlying atmosphere (*Prados-Roman et al.*, 2015, *Cuevas et al.,* 2018; *Legrand et al.,* 2018).

However, the environmental drivers of atmospheric iodine levels prior to anthropogenic influences are still unknown. Here, we show the first complete, high-resolution reconstruction of atmospheric iodine during the Holocene. We combine the ReCAP ice core measurements with other marine paleoceanographic archives to investigate the main environmental mechanisms driving millennial-scale natural atmospheric iodine variability during the Holocene.

## 2 Data and Methods

### 2.1 Study site and age model

The Renland ice cap is located on a high mountain plateau in the Scoresby Sund fjord system (coastal eastern Greenland) isolated from the main Greenland ice sheet. The 584 m long ReCAP (Renland ice CAP) ice core (71°18'18"N, 26°43'24"W; 2315 m a.s.l.) was drilled to bedrock in May-June 2015 using the Danish Hans Tausen intermediate drill system. The core is located just 2 km from the site of the original Renland ice core, drilled in 1988 to a depth of 324 m, from which a present-day accumulation rate of 50 cm ice equivalent/yr was determined (Hansson et al., 1994).

The 2015-drilled ReCAP ice core record covers the last 120 kyr BP (*Simonsen et al., 2019*). This study focuses on the upper 535 meters that constitute the Holocene period (last 11.7 kyr BP). The Holocene age-depth model is based on annual layer counting for the interval (CE 2015 - 4 kyr BP, where BP signifies 'before CE 1950'); combined with a modified

Dansgaard-Johnsen ice flow model (*Dansgaard and Johnsen*, 1969) constrained by well-dated age markers from 4 to 11.7 kyr BP. The annual layer counting was conducted using the StratiCounter algorithm (*Winstrup et al.*, 2012) constrained by volcanic eruption markers and synchronized to the GICC05 Greenland Ice Core Chronology framework (*Vinther et al.*, 2006). For details on the age model we refer the readers to Simonsen et al. (2019).

**2.2 Ice core samples and geochemical analyses**

Each one of the ice core samples devoted to the geochemical analyses (n=1035) was collected by integrating 55 cm of melted ice from a Continuous Flow Analysis (CFA) system (University of Copenhagen). The meltwater was collected in polyethylene tubes and subsequently refrozen and stored shielded from light until analyses. Samples were sent to the Environmental Analytical Chemistry laboratory (IDPA-CNR) at the University of Venice (Italy), and to Curtin University (Perth, Australia). Iodine ($^{127}$I) and sodium ($^{23}$Na) concentrations were measured by i) Sector Field Inductively Coupled Plasma Sector Field Mass Spectroscopy (ICP-SFMS, Element XR, Thermo Fisher, Germany) in Australia (defined CU system) and by ii) Inductively Collision Reaction Cell-Inductively Coupled Plasma-Mass Spectrometry (CRC-ICP-MS, Agilent 7500cx, Agilent, California, USA) in Italy (defined IDPA-CNR system). Calcium ($^{44}$Ca) was only quantified in the samples measured using the Australian system. For a full description of the analytical methodology of the sodium, iodine and calcium measurements in both laboratories, instrumental errors and detection limits, we refer the reader to Maffezzoli et al. (2018). In particular, the sodium and iodine detection limits, calculated as 3 times the standard deviation of n=80 blank values (DL=3σ), were 1 ppb and 0.005 ppb, respectively, for the IDPA-CNR system measurements and 1.1 ppb and 0.002 ppb for the CU system measurements, while the calcium measurements, only performed on the CU system, featured a detection limit of 1.6 ppb. More than 97% of all sample concentrations were above the detection limits for all 3 elements. 140 samples were analysed at both institutions for intercomparison in order to investigate differences between the analytical techniques and laboratories (Fig. S1). Although there is a non-linearity between samples with very low iodine concentrations, the ratio of the iodine measurements carried out in the two institutes average ρ = 0.95 ± 0.01 highlighting the strong correlation between the measurements at both institutes.

The ReCAP ice core temporal resolution ranged from sub-annual in the upper metres during the Great Acceleration (1950 CE- Present, average resolution of 1.31 samples/yr), subdecadal during the Late Holocene (average resolution of 0.23 samples/yr), and sub-centennial during the Neoglacial and the HTM (average resolution of 0.035 and 0.011 samples/yr respectively). The ice core iodine annual depositional mass fluxes *Iflux* were calculated according to the equation:

$$Iflux = [I]xAR$$

where (I) and AR represent the iodine concentrations and reconstructed annual accumulation rates (Fig. S2), respectively. Pearson coefficients during the main Holocene climatic phases (i.e.; HTM, Neoglacial, Late Holocene and Great Acceleration) were calculated to evaluate the statistical correlation between ReCAP geochemical variables, i.e. (I), (Na) and (Ca).

**2.3 Atmospheric Chemistry modelling**

The 1-D chemistry transport model THAMO (Tropospheric HAlogen chemistry Model) has been applied to characterize interactive halogen photochemistry and the production of tropospheric reactive iodine during the present day and three different scenarios during the Holocene (Holocene Thermal Maximum (HTM), Neoglacial and Late Holocene). THAMO has been used in the past to study halogen chemistry at different locations and environments, and further details including the full chemical scheme can be found elsewhere (*Saiz Lopez et al., 2008; Read et al., 2008; Mahajan et al., 2010; Lawler et al., 2014; Spolaor et la., 2016*). THAMO comprises 200 stacked boxes at a vertical resolution of 5 m (total height 1 km). The model includes a complete scheme for iodine, bromine, chlorine, $O_3$, $NO_x$ and $HO_x$ tropospheric chemistry, and is constrained by typical measured values of other chemical species in the marine boundary layer (MBL) (*Saiz-Lopez et al.*, 2008). The model is initialized at midnight to build up all species before the sunrise, when the sunlight starts all the photolytic processes in the gas phase. Emission fluxes of VSLs in present day scenarios are configured according to Prados-Roman et al. (*2015*). Four different scenarios have been investigated in which all of the preconditioning factors have been applied according to published paleoenvironmental reconstructions (Table 1, Fig. S3) and the MERRA reanalysis dataset (*Rienecker et al.*, 2011). Inorganic iodine emission fluxes were calculated according to laboratory-based flux parametrization (*Carpenter et al., 2013*) with mean values ranging from 1610 to 450 µg m$^{-2}$ yr$^{-1}$ for the Present-Day and the Late Holocene periods, respectively. These values were calculated using 10 ppbv of tropospheric ozone in the atmosphere before Industrialization (*Volz and Kley*, 1988) and 30 ppbv during the Great Acceleration (*Cooper et al.*, 2014). THAMO is used for a kinetic study of the impact of the different tropospheric ozone and organic iodine emission scenarios, representative of the climatic periods studied in this work, to estimate which ozone and atmospheric iodine levels best reproduce the observations from the Renland ice core. In the four different model runs (Present day, Holocene Thermal Maximum, Neoglacial and Late Holocene), we have tuned the organic iodine emissions to match the tropospheric iodine levels recorded in the ice core, given the inorganic iodine emissions determined by the tropospheric ozone concentrations, the SST and the wind speed of each scenario.

**3 Results and Discussion**

The Holocene ReCAP iodine concentrations (I) and calculated fluxes (I$_{flux}$, i.e. iodine annual depositional rates from the atmosphere to the ice surface) range from 0.06 ng g$^{-1}$ to below the detection limit (mean 0.02 ng g$^{-1}$) and from 35 to 0.01 µg m$^{-2}$ yr$^{-1}$ (mean 8.1 µg m$^{-2}$ yr$^{-1}$), respectively (Fig. 2, Fig. S2). The iodine-sodium ratio (I/Na) series shows the highest and lowest values during the HTM and the Neoglacial period respectively (Fig. S4). Iodine concentrations from ReCAP ice core may suffer from significant post-depositional processes that may affect the iodine concentration in ice at daily to seasonal scales. Nevertheless, large variations in the net iodine depositional fluxes at centennial to millennial time scales are not expected (Supplementary Information). Thus, Holocene iodine variability may respond to the interplay of different inorganic and biogenic emission sources. Laboratory studies indicate that current global inorganic iodine emissions are maintained by the reaction between iodide (I$^-$) ions with atmospheric ozone deposited to the sea surface (*Carpenter et al.*, 2013; *MacDonald et al.*, 2014). Lower inorganic iodine emissions to the atmosphere have been hypothesized (*Prados-Roman et al.*, 2015), on

account of the commensurately lower level of tropospheric ozone (about 10 ppb) before the onset of Industrialization (*Volz and Kley*, 1988). According to the THAMO model, before the present day period, the ocean inorganic iodine emissions have been low throughout the Holocene (450-509 µg m$^{-2}$ yr$^{-1}$, Table 1), contrasting the four-fold iodine concentration variability recorded in the ReCAP record during the last 11.7 kyrs BP (Fig. 2). This suggests that the Holocene high atmospheric iodine levels cannot be attributed to ozone-driven inorganic iodine emission sources. An additional source of organic iodine through the oceanic emission of VSLs is therefore needed to produce such observed variability.

### 3.1 Iodine levels during the Holocene Thermal Maximum (~11.5-5.5 kyr BP)

The ice core iodine concentrations show the highest values during the Holocene Thermal Maximum (HTM), peaking at ~10 kyr BP and remaining at the same level until ~5.5 kyr BP (mean = 0.036 ng g$^{-1}$) (Fig. 2 and S2). Similarly, the iodine fluxes remained high throughout the HTM (mean = 14.2 µg m$^{-2}$ yr$^{-1}$). This period exhibited warm surface water conditions (1.6±0.8°C higher than present (*Kaufman et al.*, 2004)) and salinity increases in the Arctic (*Briner et al.*, 2016; *Solignac et al.*, 2006) as a result of higher summer insolation (Fig. 2).

Several micropaleontological and organic geochemical biomarker records suggest a reduced sea ice coverage throughout the HTM (*Cabedo-Sanz et al.*, 2016; *Müller et al.*, 2012; *Ślubowska-Woldengen et al.*, 2008; *Vare et al.*, 2009; *Werner et al.*, 2016; *Xiao et al.*, 2017). The paleoenvironmental reconstructions carried out in multiple settings of the northern North Atlantic and the Arctic Ocean (Fig. 1, Table S1) also indicate higher primary productivity during the Early Holocene (Fig. 2). The subpolar species *T. quinqueloba*, indicative of warm and saline Atlantic Water advection (*Volkmann*, 2000), as well as of nutrient-rich subsurface waters (*Werner et al.*, 2016), was dominant in the northern North Atlantic and the Nordic Seas during the HTM, as far north as the Fram Strait (*Werner et al.*, 2013; *Werner et al.*, 2016). Higher concentrations of dinosterol and brassicasterol, respective biomarkers of dinoflagellates and diatoms (*Volkman et al.*, 1998), also indicate higher primary productivity in the subpolar North Atlantic during the HTM (*Kolling et al.*, 2017; *Müller et al.*, 2012; *Werner et al.*, 2016). The increase in nutrient supply from terrestrial sediments, derived from Arctic glacier melting and coastal erosion during the Early Holocene (*Solomina et al.*, 2015; *Wegner et al.*, 2015) may have promoted higher marine productivity in Arctic shelf and coastal areas. An increase in planktic foraminifera depositional fluxes and higher bio-induced calcium carbonate precipitation in the Northern North Atlantic and the Nordic Seas (*Telesiński et al.*, 2015; *Werner et al.*, 2013) (Fig. 2, Table S1) provide further evidence of enhanced primary productivity. These biological species, including phytoplankton and macro- and micro-algae, accumulate iodine to concentrations of up to $10^3$-$10^6$ greater than in seawater and produce iodine compounds in the ocean (*Saiz-Lopez et al.*, 2012b). An increase in primary productivity would lead to an enhancement of the biological iodine production in the ocean, followed by its release to the atmosphere resulting in what could be defined as a "Biological Iodine Explosion" (BIE) (Fig. 3). Maximum summertime solar irradiance in the Arctic during the HTM (Fig. 2) would also slightly increase the algae oxidative stress increasing the iodine effusion rates from the surface ocean (*Saiz-Lopez et al.*, 2012b; Saiz-lopez; 2015). Additionally, higher sea surface temperatures (SST) during the HTM would favour the sea-air phase transfer of volatiles (i.e. iodine compounds) produced at the ocean surface. Greater primary productivity in the

reduced summertime sea ice-covered Nordic Seas drives two biological changes resulting in enhanced iodine release from the ocean surface: firstly, enhanced emission of iodocarbons from more extended open ocean directly to the atmosphere and, secondly, the enhanced production of iodine ($I_2$) by sea-ice phytoplankton colonies during spring and its release to the atmosphere through brine channels and ice cracks formed in the thinner HTM seasonal sea ice (*Saiz-Lopez et al.*, 2015) (Fig. 3). Inorganic iodine emissions (~509 µg m$^{-2}$ yr$^{-1}$) would only account for 27% of the total HTM iodine emissions, as inferred by the THAMO model (Table 1). Therefore, the additional 73% percent must be supplied by organic iodine sources to maintain the high levels of tropospheric iodine recorded in the ice core. This amount of organic sources (1368 µg m$^{-2}$ yr$^{-1}$, Table 1) is twice the amount necessary to model the present day scenario, in which the high contribution (~70%) of inorganic sources (1610 µg m$^{-2}$ yr$^{-1}$, Table 1) suggests that only 684 µg m$^{-2}$ yr$^{-1}$ are necessary to model the present day tropospheric iodine levels. No significant correlation is found between iodine and sodium or calcium ice core concentrations (Table S2), well-recognized proxies of ssa and atmospheric dust in Greenland respectively (*Schüpbach et al., 2018*). Thus, the lack of correlation between iodine and calcium suggests that atmospheric transport of ssa or to dust-related particle-bound iodine are minor contributors explaining the iodine variability during the HTM. We therefore conclude that, according to our modelling results, ocean biogenic iodine production and emission played the dominant role (up to 73% of iodine emissions, Table 1) in modulating the observed millennial-scale iodine record during the HTM period (~11.5-5.5 kyr BP).

**3.2 Iodine levels during the Neoglacial Period (~5.5-3.4 kyr BP)**

An abrupt decrease in iodine levels occurs during the Neoglacial period (Fig. 2 and S1), a time interval characterized by a general cooling of the Arctic and North Atlantic regions (*Jennings et al.*, 2002; *Koç et al.*, 1993; *Nesje and Dahl*, 1993). Iodine concentrations and fluxes are at their lowest levels in the entire sequence. A significant correlation exists between iodine and sodium (Table S2) suggesting an influence of ssa variability on iodine concentrations during the Neoglacial period.

According to the iodine series, the HTM ended quite abruptly *ca*. 5.5 kyr BP when global sea level reached its modern level and a hence intensified sea ice production in the Kara and Laptev Seas (Russian Arctic) resulted in a perennial sea ice cover in the central Arctic Ocean and enhanced sea ice export into the North Atlantic (*Bauch et al.*, 2001; *Cronin et al.*, 2010; *Telesiński et al.*, 2015; *Werner et al.*, 2013). Increased sea ice coverage and/or reduced summer sea ice break up is also reported for the Greenland-Norwegian Sea and eastern Fram Strait after c. 5.5 kyr BP (*Koç et al.*, 1993; *Müller et al.*, 2012; *Telesinski et al.*, 2014a; *Telesiński et al.*, 2014b; *Werner et al.*, 2013). Recent sea-ice reconstructions from the North Iceland Shelf show that the HTM was followed by a pronounced re-advance of sea ice extent and thickness at about 6 kyr BP (*Cabedo-Sanz and Belt*, 2016; *Xiao et al.*, 2017), when SST started to decrease (*Kristjánsdóttir et al.*, 2017). Increased sea-ice cover ultimately reduces atmospheric iodine emissions through a number of direct and indirect processes involving reduced marine primary production (Fig. 2; Table S1 and references therein). Thicker sea ice (~ 3m) would reduce the amount of sunlight reaching algal colonies underneath sea ice, thus reducing the amount of seawater iodine compounds (*Saiz-Lopez et al.*, 2015). Thicker and/or more extensive sea ice would also impede the iodine diffusion through ice and further release of $I_2$ to the overlying

atmosphere (*Saiz-Lopez et al.*, 2015). Both of these processes likely contributed to the observed ice core iodine minimum during the Neoglacial period with mean iodine concentrations and fluxes of 0.008 ng g$^{-1}$ and 3.9 µg m$^{-2}$ yr$^{-1}$, respectively (Figs. 2 and 3). Our modelling results indicate that inorganic iodine emissions (478 µg m$^{-2}$ yr$^{-1}$) account for most of the low
atmospheric iodine concentrations during the Neoglacial period (Table 1), although a 15% of atmospheric iodine still originate from organic sources (85.5 µg m$^{-2}$ yr$^{-1}$). In this scenario the low inorganic emissions caused by low tropospheric ozone, in addition to the low organic emissions due to thicker sea ice would lead to the lowest modelled levels of tropospheric iodine of the entire Holocene period.

### 3.3 Iodine levels during the Late Holocene (~3.4 kyr BP- Present)

Iodine levels vary widely during the Late Holocene, with concentration and fluxes ranging from 0.064 ng g$^{-1}$ to below the detection limit (mean 0.017 ng g$^{-1}$) and from 35 to 0.015 µg m$^{-2}$ yr$^{-1}$ (mean 8 ug m$^{-2}$ yr$^{-1}$), respectively. Iodine concentrations slightly increased at ca. 3.4 kyr BP and remained fairly constant until 0.8 kyr BP (mean concentrations and fluxes of 0.014 and 6.34 ug m$^{-2}$ yr$^{-1}$ respectively) (Fig 2 and S2).

Arctic sea ice extent progressively increased during the Late Holocene as recorded in mid- and high-latitude Arctic locations (e.g. *Cabedo-Sanz et al.*, 2016; *Werner et al.*, 2013; *Werner et al.*, 2016) (Fig. 2, Table 1). Sea ice reconstructions from the East Greenland shelf, however, reveal a slightly reduced but stable sea ice cover during the last 3 kyr consistent with increasing phytoplankton growth and marine productivity (*Kolling et al.*, 2017) (Figs. S5 and S6). This trend is consistent with moderately increasing iodine levels recorded in the ReCAP ice core over the last 3 kyr (Figs. 2, S5 and S6).

Ice core iodine concentrations and fluxes increased by 50% during the last eight centuries reaching mean values of ~0.021 ng g$^{-1}$ and 8.9 ug m$^{-2}$ yr$^{-1}$, respectively, at the onset of the Industrial Period (CE 1750-1850). The iodine time series in this part of the record reveals a higher frequency variability mainly due to a reduced ice compression towards the surface and thus an increased temporal resolution of the measurements. Model results indicate that organic and inorganic iodine emissions contributed almost equally (43% and 57%, respectively; Table 1) to the total iodine released to the atmosphere during the Late
Holocene. Phytoplankton productivity in the Eastern Greenland shelf varied significantly during the Little Ice Age (LIA, CE 1400-1850), even reaching the highest values since the HTM (*Kolling et al.*, 2017) (Fig. 2). Sea ice re-advances and/or intensified sea ice export from the Arctic Ocean along the Greenland shelf and North of Iceland during the LIA (*Cabedo-Sanz et al.*, 2016; *Kolling et al.*, 2017) and a 10% decrease in solar irradiance since the Early Holocene (Fig. 2; Table S1) likely explained the atmospheric iodine levels being lower in the late Holocene with respect to the higher levels recorded during the
HTM.

### 4 Conclusions

Iodine levels in the ReCAP ice core have tripled since the onset of the Great Acceleration (1950 CE- Present) (*Cuevas et al.*, 2018). This very recent atmospheric iodine increase has been explained by the rise in anthropogenic ozone pollution since the mid-20[th] century (*Cooper et al.*, 2014; *Cuevas et al.*, 2018; *Legrand et al.,* 2018), since oceanic inorganic emissions of HOI

and I$_2$ depend on tropospheric ozone concentrations (*Carpenter et al.*, 2013; *MacDonald et al.*, 2014; *Prados-Roman et al.*, 2015). A recent global modelling study has shown that inorganic iodine emissions from the oceans constitute ~75% of present-day total atmospheric iodine (*Prados-Roman et al.*, 2015). In the absence of human disturbances to the iodine biogeochemical cycling during the HTM, the remarkably high iodine levels recorded in the ReCAP ice core can only be explained by natural drivers influencing iodine emissions to the atmosphere. Our results show that at the onset of the Holocene, enhanced ocean primary production coupled with maxima in solar irradiance and open water conditions in the Arctic Ocean and in the Nordic Seas (*Bauch et al.*, 2001; *Cronin et al.*, 2010; *Müller et al.*, 2012; *Telesiński et al.*, 2014; *Werner et al.*, 2013; 2016) controlled iodine emissions to the atmosphere (Fig. 3), resulting in a four millennia-long period of high atmospheric iodine concentrations. The decrease in iodine levels observed at the onset of the Neoglacial period coincides with environmental modifications in the Arctic, primarily the advance of sea ice and the reduction of marine primary production (Fig. 2, Table 1 and references therein).

The fluctuations in atmospheric iodine levels recorded in the ReCAP ice core most likely resulted in significant long-term environmental impacts that ultimately affected the Holocene atmospheric chemistry and associated radiative impacts, such as ozone depletion (*Saiz-Lopez et al., 2012a; Hossaini et al., 2015; Sherwen et al., 2017*) and new particle formation (*Allan et al.,2015; Roscoe et al., 2015, Sipilä et al., 2016*). The Early Holocene high iodine emissions might have led to enhanced aerosol concentrations in the Arctic atmosphere since atmospheric iodine promotes the formation of new ultrafine aerosol particles in coastal (*Mahajan et al.*, 2011; *O'dowd et al.*, 2002; *Sipilä et al.*, 2016) and polar regions (*Allan et al.*, 2015; *Roscoe et al.*, 2015). Furthermore, higher iodine levels have the potential to accelerate considerably atmospheric tropospheric ozone loss by up to 14 % and 27 % in the global marine boundary layer and upper troposphere, respectively (*Saiz-Lopez et al.*, 2014). Tropospheric ozone loss ultimately affects the oxidative capacity of the atmosphere and ozone radiative forcing which, in turn, can impact climate by cooling the global troposphere by approximate -0.1 W m$^{-2}$ (*Hossaini et al.*, 2015; *Saiz-Lopez et al.*, 2012a; *Sherwen et al.,* 2017). The high variability of marine iodine fluxes to the continents could also have had implications for land ecosystems during the Holocene since iodine is a key trace element in animal and human endocrine systems *(De Benoist et al., 2004)*.

Our results highlight that the increase in atmospheric iodine levels since 1950 CE is neither acute nor unusual in the context of long-term (i.e. millennial-scale) iodine variability. Therefore, the high levels of atmospheric iodine which occurred during the early Holocene may serve as an analogue for future atmospheric composition and climate conditions. This is particularly relevant to the Arctic, for which ice-free summertime conditions have been forecasted to occur by 2050 CE (*Overland and Wang*, 2013).

**5 Data availability**

The ice core and model data that support the findings of this study will be made available on the PANGAEA and NOAA paleoclimate public databases after publication. Further complementary dataset could be available upon request.

## 6 Author contribution

A.S.L., P.V. and A.S. designed the research. J.P.C., N.M., H.A.K., C.A.C., G.C., C.B., P.V. and R.E. collected and measured the samples, and analyzed the resulting data. B.V. constructed the chronology. J.M. supplied paleo primary production and sea ice coverage data. All authors interpreted the results. J.P.C. and A.S.L. wrote the manuscript with contributions from all authors.

## 7 Competing interests

The authors declare no competing interests.

## 8 Acknowledgments

This work was supported by CSIC. The RECAP ice coring effort was financed by the Danish Research Council through a Sapere Aude grant, the NSF through the Division of Polar Programs, the Alfred Wegener Institute, and the European Research Council under the European Community's Seventh Framework Programme (FP7/2007-2013) / ERC grant agreement 610055 through the Ice2Ice project and the Early Human Impact project (267696). JPC held a Juan de la Cierva – Incorporación postdoctoral contract (ref. IJCI-2015-23839). J.M. received funding through a Helmholtz Research grant VH-NG-1101. This study has received funding from the European Research Council Executive Agency under the European Union´s Horizon 2020 Research and Innovation programme (Project 'ERC-2016-COG 726349 CLIMAHAL').

## 9 Tables

| | I ug/g ReCAP | SST/ºC | [iodide] | [O3] | Wind speed (m s$^{-1}$) | HOI flux ug m$^{-2}$ y$^{-1}$ | I$_2$ flux ug m$^{-2}$ y$^{-1}$ | Total_inorg I flux ug m$^{-2}$ y$^{-1}$ | Total_org ug m$^{-2}$ |
|---|---|---|---|---|---|---|---|---|---|
| HTM | 0,036 | 8.85 | 1,251E-08 | 10[*] | 7 | 486 | 23 | 509 | 1368 |
| Neoglacial | 0,008 | 8.57 | 1,212E-08 | 10[*] | 7 | 457 | 22 | 478 | 85.5 |
| Late Holocene | 0,017 | 8.3 | 1,175E-08 | 10[*] | 7 | 429 | 21 | 450 | 342 |
| Present day | 0,038 | 9.1 | 1,288E-08 | 30[**] | 7 | 1539 | 72 | 1610 | 684 |

**Table 1: Iodine concentrations in the ReCAP ice core and THAMO modelled oceanic inorganic and organic iodine emission fluxes during the Present Day and the different Holocene scenarios**. Inorganic (HOI and I$_2$) and organic (VSLs) iodine fluxes have been configured according to MacDonald et al., 2014 and Prados-Roman et al., 2015. Total inorganic iodine flux is the sum of HOI and I$_2$ fluxes. Total organic iodine flux comprises the fluxes $CH_3I+CH_2I_2+CH_2IBr+CH_2ICl$. *Values obtained from Volz and Kely (1988). **Values obtained from MERRA reanalysis dataset.

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

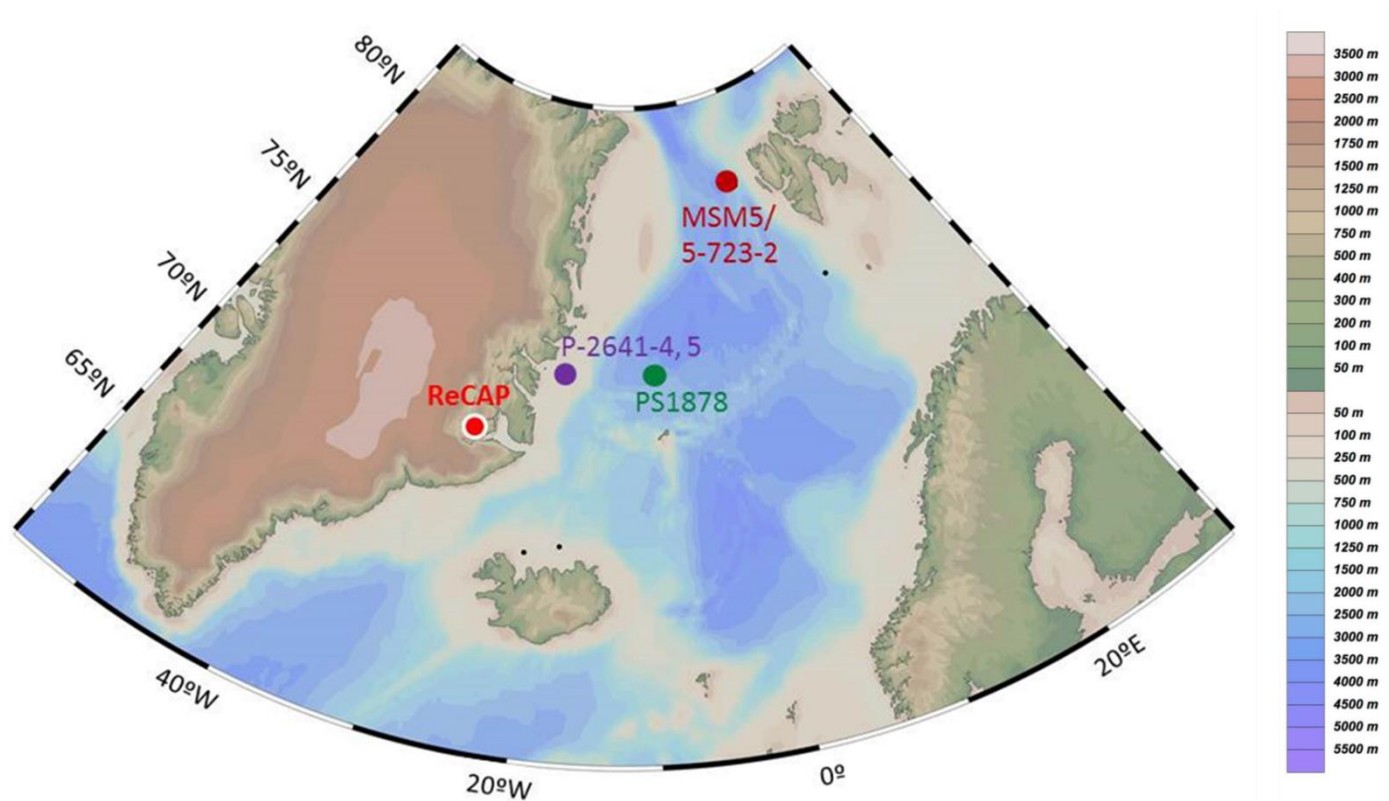

**Fig. 1: Location of the ReCAP ice core (red) and other marine paleoceanographic archives in the Nordic Seas discussed in the text**.
For references of the paleoenvironmental proxies from the displayed sediment cores, please refer to the text and Fig. 2 caption.
Core PS1878 (*Telesiński et al.*, 2015); Core PS2641-4 (*Müller et al.*, 2012); Core MSM5 5/723-2 (*Werner et al.*, 2013; *Werner et al.*, 2016).
Bathymetric map obtained from Ocean Data View software (Sclitzer, 2015). Bathymetry is shown in the colour bar

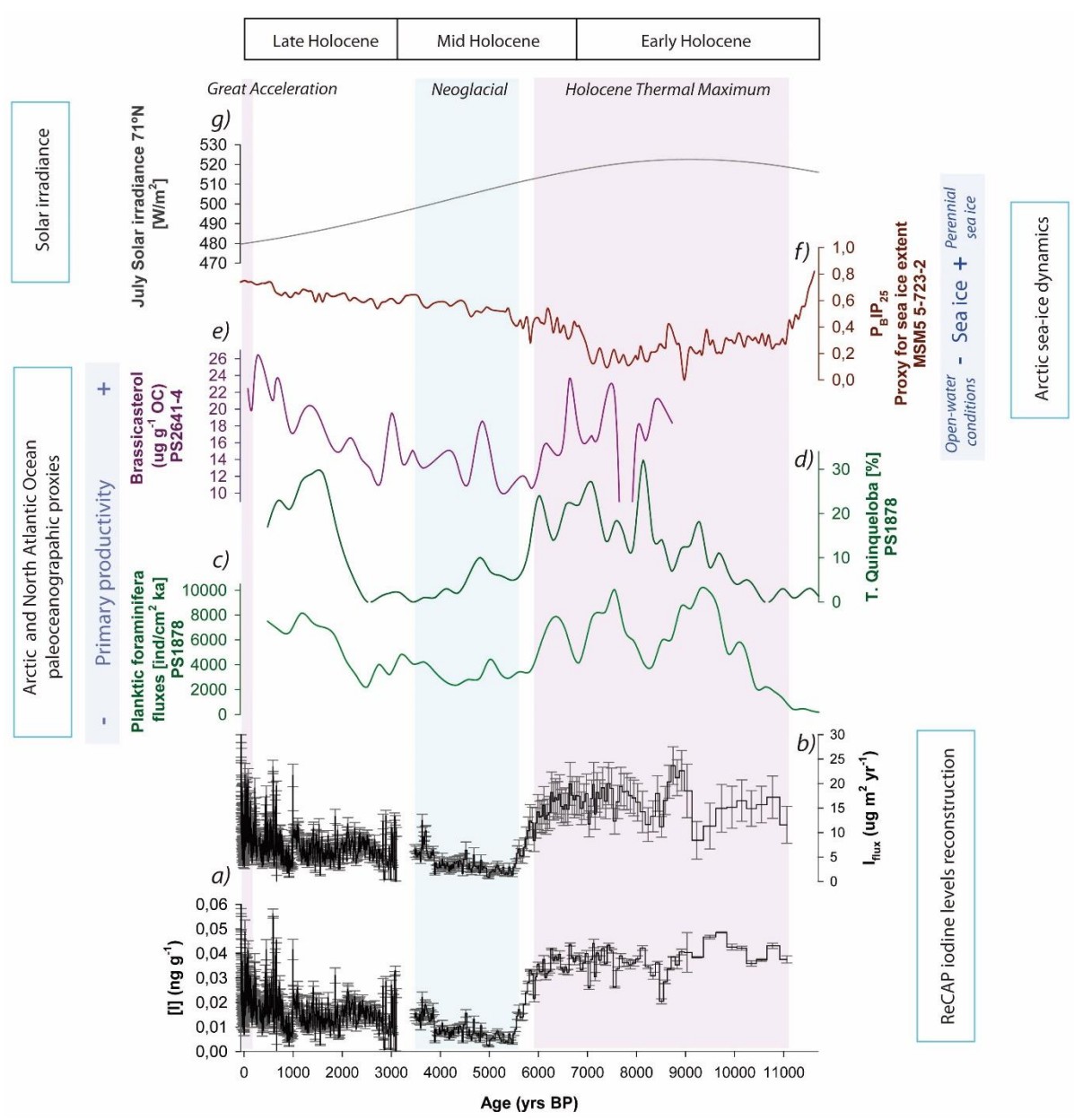

**Fig. 2: Holocene iodine concentrations and fluxes evolution from the ReCAP ice core and primary productivity and sea ice proxies from the Nordic Seas.** From bottom to top: **a)** Iodine concentrations (1σ, experimental uncertainties; (Iodine detection limits are within 0.005 and 0.002 ppb) and **b)** Iodine fluxes (1σ, propagated from the concentration and accumulation rate uncertainties) (N=1050); **c)** and **d)** 540    Planktic foraminifera and *T. quinqueloba* (core PS1878) (*Telesiński et al.*, 2015), **e)** *Brassicasterol* (core PS2641-4) (*Müller et al.*, 2012); **f)**

Sea ice cover (core MSM5/5-723-2) (*Werner et al.*, 2013; *Werner et al.*, 2016); **g)** 71ºN July solar irradiance. Color boxes indicate the Holocene main climatic periods mentioned in the text; pink boxes indicate warmer phases while blue boxes indicate colder intervals.

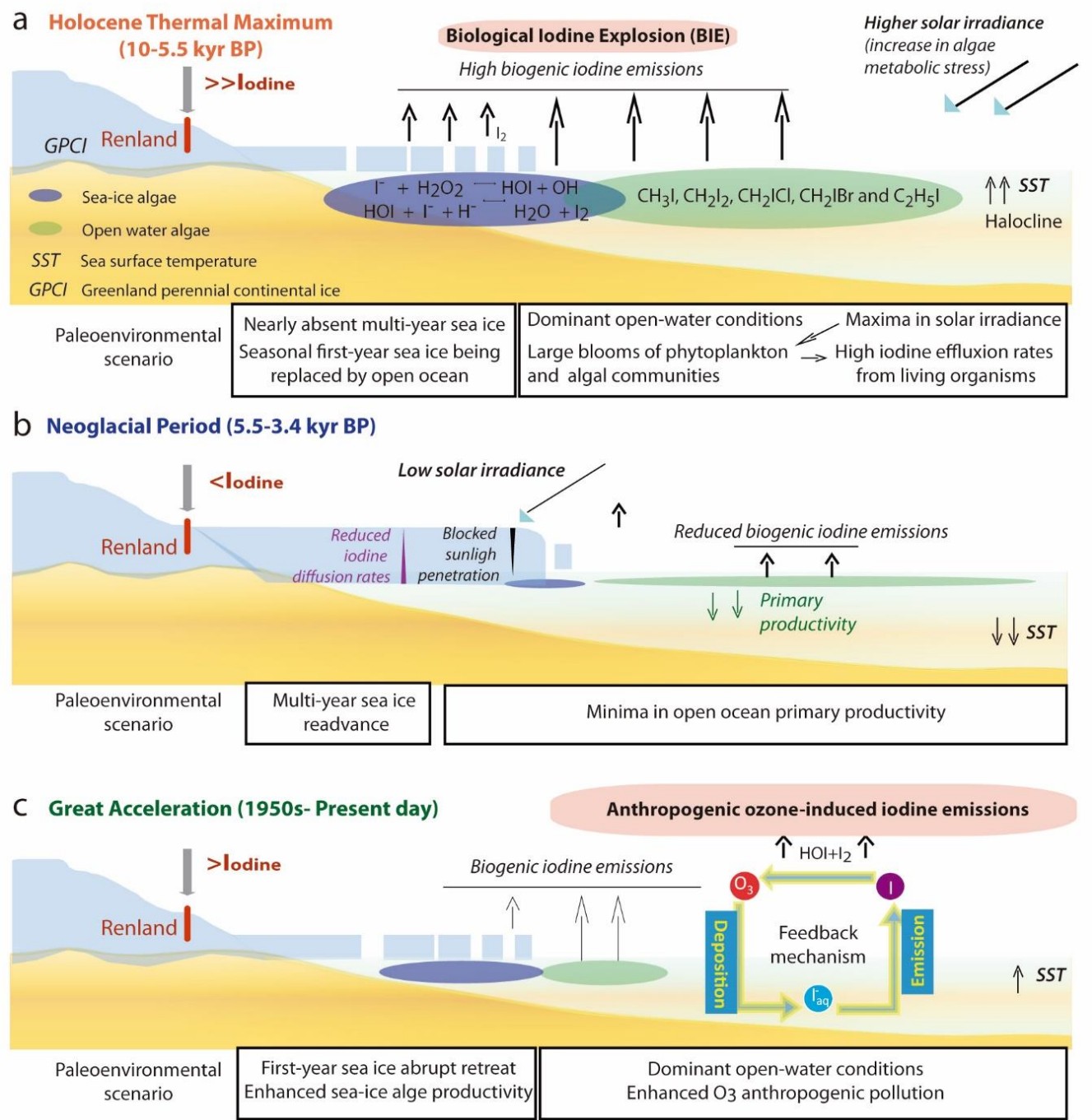

**Fig. 3: Schematic diagram showing the coupled ocean-atmosphere iodine biogeochemical cycle throughout the Holocene.** (a) Holocene Thermal Maximum. (b) Neoglacial period. (c) Great Acceleration.