# Peer review of "Holocene atmospheric iodine evolution over the North Atlantic"

_Climate of the Past, 2019_

## Referee Comment (RC1) · Anonymous Referee #1 · 29 Jul 2019

Review of Corella et al. (https://doi.org/10.5194/cp-2019-71)

General comments Corella et al. present a relatively high-resolution ice core record of atmospheric iodine variability during the Holocene from the Renland Ice Core, Greenland. Iodine concentrations are extremely low in ice cores (<0.1 ng g-1), and require meticulous sample preparation and ultra-trace analytical protocols. This study presents a very nice lab intercalibration experiment between two ice core labs - Curtin University, Australia and IDPA-CNR, Italy, which I commend. Over the Holocene, the authors suggest the large variability in iodine concentrations and iodine fluxes, measured in the ice core, reflect changes in marine primary production over the North Atlantic. They derive a conceptual model to explain the three-step changes in iodine levels over the last 11 ka, i.e., Holocene Thermal Maximum, Neoglacial Period, and Late Holocene.

[Figure]

This iodine record will be of interest to a number of communities including atmospheric chemists, palaeoclimatologists and marine biogeochemists. However, the authors have not addressed post-depositional processes that are known to modify the original iodine deposition to snow. These photochemical processes need to be quantified before ice core records of iodine can be interpreted at the Renland site. After addressing the issue of post-depositional processes, I am happy to recommend this manuscript for publication in Climate of the Past.

Specific comments Post-depositional processes, such as UV-photolysis, can cause reactive halogen species in the snowpack to be lost to the overlying atmosphere (e.g. Frieß et al., 2010;Gálvez et al., 2016;Simpson et al., 2005), i.e., the snowpack can release gas-phase iodine. Firstly, this local iodine source needs to be added in the "origins and cycling of iodine" section in the introduction. Secondly, the manuscript is lacking discussion on photochemical post-depositional processes related to iodine loss from the snowpack. Such processes need to be quantified, especially at individual ice core sites due to their relationship with snow accumulation, before archived concentrations of iodine can be interpreted. Do you have any surface snow and atmospheric iodine measurements from Renland you could use? At the very least, you could make some assumptions that post-depositional processes are negligible at relatively high accumulation sites, such as Renland, but this would need to be backed up by evidence from the literature.

The authors have produced a nice schematic diagram (Figure 3) showing the three phases of iodine evolution over the Holocene. The figure is only mentioned once in text and I recommend a fuller explanation of the figure throughout the discussion.

The study uses a combination of ice core and modelling evidence. The manuscript is rather ice core heavy and I recommend extending the modelling results and discussion sections especially for ice core readers who are not experts in atmospheric chemistry.

Lastly, I suggest restructuring the manuscript to combine the results and discussion

into one section, and making the final conclusions separate. That would leave the reader with the key take home messages of the study.

Technical corrections L17 Insert "the" between "large influence on. . .oxidising capacity" L21 Spell out "ReCAP" acronym L21 Which region does your atmospheric iodine represent? L23 Delete "-" between "before. . .present" L23 Replace "ocean" with "marine" L24 "Biological iodine explosion" is this term reported previously or are you suggesting it? Needs a little further explanation. L24 What is the "iodine trend" doing during the early Holocene? L29 This sentence is out of place. The paragraph is about the impact of halogen chemistry on the oxidising capacity of the atmosphere and thus your topic sentence should reflect this. L30 Remind the reader what time period the Holocene encompasses. L32 Please add reference. L45 Throughout the manuscript, you refer to ozone as "ozone", "tropospheric ozone", "atmospheric ozone", "surface ozone". Please be consistent with your terminology. L51 Here you need a greater explanation of the recycling processes of iodine in surface snow. L58 Capital "Ice". L60 You states there are only two iodine records from ice cores. However, later in the paragraph you mention other records. Please add references for all iodine ice core records (e.g. Spolaor et al., 2013;Cuevas et al., 2018;Legrand et al., 2018). L66 What does the Talos Dome iodine record reflect? L72 Add the reference for the "recent study" upfront. L86 Add "the" between "plateau and. . .Scoresby" L91 Please add reference for ReCAP age model. L101 Were samples transported frozen? L102 Replace "of" with "at" L102 "Curtin University of Technology" is now "Curtin University". Please replace here and throughout the manuscript. See https://www.curtin.edu.au/ L105 Change "CUT" to "CU" here and throughout. L105 Please add reference to the Curtin University ICP-MS method or if these are the first measurements for iodine reported from that lab then please add methods and a table in the supplement with the operating parameters (e.g. torch, spray chamber, nebuliser). L106 You define the Italian ICP-MS as "IDPA-CNR" please make sure you are consistent with your terminology throughout the manuscript. L107 Reference is repeated twice. Please delete one. L107 Add resolution for sodium measurements. L109 Spell out ultrapure water "UPW" acronym.

L109 What are these instrumental errors? Are they related to reproducibility or accuracy? Please add accuracy and precision for iodine and sodium measurements. What certified reference material did you use? L110-111 Use consistent terminology for Italian and Australian ICP-MS. L113 Yes there is a strong correlation between the two labs but it is non-linear at low concentrations. Include a note on the large area at low iodine concentrations. L116 Add "our" between" model and…sampling". L116 Add "mass" between "depositional…fluxes". L118 Please add reference for accumulation rates. L119 Please add the different climatic phases in the introduction. L121 Calcium is not mentioned previously. Either include Ca methods or remove entirely. L124 MBL spell out acronym. L128 First mention of photolytic processes. Need to include iodine snow photochemistry in the introduction. L132 HTM spell out acronym. L132 Delete "rs" in ""ky-1". L135 Replace "results" with "ice core and modelling results" and add a fuller explanation of the modelling results here. Could you add a figure or table summarising the results? L137-138 Symbol $\mu$. L139 What causes re-mobilisation? L140 Please add reference of iodine concentration and accumulation rate studies. You haven't quantified how the accumulation rate impacts iodine loss at the Reland site so please include some evidence to justify the your assumption that post-depositional processes are negligible. L154 Symbol $\mu$. L147 What are the biomarkers? L166 "The increase in nutrient supply from terrestrial sediment delivery". L172 Replace "The" with "An". L174 Can you provide a reference "biological iodine explosion". L184 ssa acronym. L185 Please mention calcium dust proxy and add reference. L185-186 Is this result from your model? L193 Replace "on" with "of". L186 Period. L194 Replace "value" with "level". L195 Add location of Kara and Laptev Seas to Fig. 1. L200 What measure of sea ice referring to? Sea ice extent or concentration or thickness? Here and throughout manuscript please specify. L204 How much thicker does the sea ice need to be? Can you give estimates of thickness and light penetration depth? L204 Add sea ice as a source of iodine in the introduction. L211 Should the fluxes and concentrations be reported as 2 significant figures? L222 Can you quantify the frequency? L223 What is the second part? L247 What are the associated radiative impacts? Figure 1 Mention the references in the caption to help the reader. Figure S1 What does the 1:1 line indicate? Be consistent with the naming of the Australian and Italian iodine measurements.

References Cuevas, C. A., Maffezzoli, N., Corella, J. P., Spolaor, A., Vallelonga, P., Kjær, H. A., Simonsen, M., Winstrup, M., Vinther, B., and Horvat, C.: Rapid increase in atmospheric iodine levels in the North Atlantic since the mid-20th century, Nature communications, 9, 1452, 2018. Frieß, U., Deutschmann, T., Gilfedder, B., Weller, R., and Platt, U.: Iodine monoxide in the Antarctic snowpack, Atmospheric Chemistry and Physics, 10, 2439-2456, 2010. Gálvez, Ó., Baeza-Romero, M. T., Sanz, M., and Saiz-Lopez, A.: Photolysis of frozen iodate salts as a source of active iodine in the polar environment, 2016. Legrand, M., McConnell, J. R., Preunkert, S., Arienzo, M., Chellman, N., Gleason, K., Sherwen, T., Evans, M. J., and Carpenter, L. J.: Alpine ice evidence of a three-fold increase in atmospheric iodine deposition since 1950 in Europe due to increasing oceanic emissions, Proceedings of the National Academy of Sciences, 115, 12136-12141, 2018. Simpson, W. R., Alvarez‐Aviles, L., Douglas, T. A., Sturm, M., and Domine, F.: Halogens in the coastal snow pack near Barrow, Alaska: Evidence for active bromine air‐snow chemistry during springtime, Geophysical research letters, 32, 2005. Spolaor, A., Vallelonga, P., Plane, J., Kehrwald, N., Gabrieli, J., Varin, C., Turetta, C., Cozzi, G., Kumar, R., and Boutron, C.: Halogen species record Antarctic sea ice extent over glacial–interglacial periods, Atmospheric Chemistry and Physics, 13, 6623-6635, 2013.

---

## Referee Comment (RC2) · Anonymous Referee #2 · 12 Aug 2019

In this study, the first ice core iodine record spanning the Holocene is presented. Iodine levels were high during the early Holocene and industrialization relative to the late Holocene. Chemical transport model results of inorganic iodine sources and transport are compared to ice core iodine levels. This comparison suggests that marine biogenic sources of iodine were higher during the early Holocene relative to the late Holocene. This increase in biogenic sources is reasonable in the early Holocene given that other reconstructions show that sea surface temperatures in the North Atlantic were warmer, salinity was higher, sea ice extent was lower, and primary productivity of subpolar planktonic species was higher. This study is well-written and well-organized. It should be published because it is the first record iodine covering the early Holocene and provides strong links to climatic factors that could explain the unexpectedly high

levels of early Holocene iodine. Hopefully, it will inspire future study of iodine species in ice cores.

**Major comments**

Section 1, lines 36-37: This sentence is confusing. What exactly is meant by "has a global contribution of up to 27% of the total rate of ozone loss?"

Section 1, lines 58-68: Two ice cores are reported as having the only iodine records, but further in the paragraph, a third ice core record is discussed. Please clarify.

Section 2.1, line 91-96. Please discuss the depth-age scale in more detail because (it is not yet published?). Perhaps include a depth-age scale in the supplement. Which volcanic markers were used and how well-spaced are they throughout the record? What are the uncertainties?

Section 2.2, lines 99-100. Please include the sampling resolution at various depths. Clearly, several more samples were analysed in more recent years which makes sense as the fractions were collected via the CFA system. The easiest way to do this may be to include a mean sampling resolution for the time periods discussed in each section (results).

Section 3: Please use the same units when possible. It is difficult for the reader to understand readily the comparisons between different periods of time and the model versus the observations when the units are changed.

Section 3: It would be useful to discuss the relationship between concentration and flux here. Did the flux calculation confirm that remobilization processes were not consequential? Are there any important differences when flux is used?

Line 141: Start a new paragraph here? How does the second part of this paragraph relate to the first part?

Correlation coefficients of sodium and calcium are shown in Table S3 and in section
3.1, lines 182-183 the authors state that there is no correlation. Is it possible to also include a time series of calcium and sodium measurements in the supplement?

Lines 191: Please use the same units throughout the manuscript when possible.

Lines 192-193: How would the ssa contribution change over time? Is the lack of a correlation during the early Holocene purely due to the biogenic contribution overwhelming the other signals? Are they still there?

Lines 205-208: Please include the ice core values here and clarify how this conclusion was made, given the values provided.

Lines 221-223: Replace "a higher frequency" with "higher frequency variability." I agree that this is pretty clearly due to increased sampling resolution. If anything, averaging or smoothing could be used to compare the late Holocene to the early Holocene. The late Holocene levels may only appear to show higher frequency variability because there are more samples.

The focus of this study is iodine and should remain iodine. Do other halogens, like bromine, that were measured in the ReCAP ice core (Maffezzoli et al., 2018, in review), also show the same high levels during the early Holocene? Are the processes governing the sources, transport, and deposition of other halogens so different that they should not be included in the supplement?

Consider adding a section about the modelling results. These results are used to explain the interpretation of the ice core record, but it would be useful to have an understanding of the results prior to the comparison.

Figures and tables

fig. 1 add mapping software reference and color bar if going to include bathymetry fig. 2 add indication of detection limits to iodine concentrations Table S1: Please consider moving table S1 to the main text. Please add more to this caption. What are the sources (ice core versus model inputs/outputs) in each column?

CPD
Minor Comments: Abstract, line 22: change to: "in the record were found" line 30: "allows for the detailed" line 32: "key factor for understanding" line 42: change to "involve" line 59: "Ice core" line 66: "allow for a" Rename section 2.1 "Study site and dating" or "Study site and age scale" Line 108: "with the stability of the instrumental" Line 126: remove "The model" Line 129: "all of the" Line 138: add comma before "respectively" Line 141: remove "thus" before "Holocene" Line 146: remove "occurring" before "before" Line 203: change to "algal" Line 205: change to "both of these processes" Line 207, 220: "concentrations" Line 221: commas before and after "respectively" Line 241: comma before "resulting" Line 251: "to accelerate considerably"

---

## Referee Comment (RC3) · Anonymous Referee #3 · 13 Aug 2019

In the paper "Holocene atmospheric iodine evolution over the North Atlantic" by J. P. Corella and co-authors a new record of iodine concentration in the ReCAP ice core (Greenland) is presented. The record covers the full Holocene and an interesting discussion about iodine variations in the atmosphere across different climatic periods is shown. In particular the authors elaborate here an easy conceptual model which seems to be very effective in explaining iodine variability during the three time periods here considered: Holocene Thermal Maximum, Neoglacial period and Great Acceleration. The paper shows a very good laboratory intercomparison exercise, showing a general agreement in the measured iodine concentrations. This work is generally well presented and well written and will surely be a milestone in the use of halogens measured in ice cores to achieve information about Sea Ice in the North Atlantic. My main

concern about the paper is the lack of discussion about the possible post-depositional effects of iodine species in the snow. A detailed discussion about the preservation of iodine compounds in the snow and their stability in time due to a possible photochemical degradation should be added to the text. Before concluding that iodine is a real marker reflecting the past North Atlantic conditions, the authors should assess that the very low concentrations they are measuring are really telling us a story about past conditions, or, at least, that the record obtained is not significantly different from what was deposited in the Renland Ice Cap. Moreover, the meteorological conditions were surely very different in the different time periods here shown, and the authors should clarify why the variability of iodine is mainly linked to the primary productivity and not to changes in transport efficiency through time. Other papers by the authors reported an halogen enrichment factor by subtracting the sea salt contribution. I'm not sure about which is the best way to proceed, but I would like to know why in this paper this elaboration was not considered. I suspect that an elaboration of the iodine values with sodium can infer to iodine a variability which mainly comes from Na+ concentrations. Once this points will be addressed I would recommend the publication of this paper.

Minor issues.

Line 59: Change to "Ice core..." Line 109: explain what UPW means. Line 110: How were the d.l. calculated? Explain briefly how the two labs measured this figure of merit of the two methods. Line 116: Is there an age model for the RECAP ice core? Give more information on this point: how was the age scale built? Which absolute markers were used (i.e. volcanoes, tephra, etc)? Which is the uncertainty in the bottom part of the ice core? Line 137-138: units are not readable... Line 141: remove the comma between m-2 and yr-1. Line 198: Two references by Telesinski et al. were published in 2014, please call them a) and b) in order to make clear which reference you are referring to in the text. Line 198: is c. for ca.? Line 240: remove M.M. before Telesinski. Line 412: check the references: something wrong happened to the titel of this paper. Figure 2 caption: use "concentration and flux" or "concentrations and fluxes", not a mix.

Figure 2: during the HTM the authors' reconstruction seems to be very smoothed with respect to the marine records. Is it due to the poor resolution at this depth? Please discuss this point in the text. Figure 2 and text: is flux adding something new with respect to concentration? The two profiles are very similar and the authors should discuss this point in terms of wet/dry deposition mechanisms at the drilling site.

[Figure]

---

## Author Comment (AC1) · 26 Sep 2019

Referee #1: General comments Corella et al. present a relatively high-resolution ice core record of atmospheric iodine variability during the Holocene from the Renland Ice Core, Greenland. Iodine concentrations are extremely low in ice cores (<0.1 ng g-1), and require meticulous sample preparation and ultra-trace analytical protocols. This study presents a very nice lab intercalibration experiment between two ice core labs - Curtin University, Australia and IDPA-CNR, Italy, which I commend. Over the Holocene, the authors suggest the large variability in iodine concentrations and iodine fluxes, measured in the ice core, reflect changes in marine primary production over the North Atlantic. They derive a conceptual model to explain the three-step changes in iodine levels over the last 11 ka, i.e., Holocene Thermal Maximum, Neoglacial Period,

and Late Holocene. This iodine record will be of interest to a number of communities including atmospheric chemists, palaeoclimatologists and marine biogeochemists. However, the authors have not addressed post-depositional processes that are known to modify the original iodine deposition to snow. These photochemical processes need to be quantified before ice core records of iodine can be interpreted at the Renland site. After addressing the issue of post-depositional processes, I am happy to recommend this manuscript for publication in Climate of the Past.

Response: We appreciate the positive feedback from reviewer 1. We have addressed below all the comments and suggestions made by reviewer that have certainly improved the quality of the manuscript. In particular, we have added a new section in the revised manuscript in which we address all the possible post-depositional processes occurring at the Renland site.

Specific comments Referee #1:Post-depositional processes, such as UV-photolysis, can cause re-active halogen species in the snowpack to be lost to the overlying atmosphere (e.g.Frieß et al., 2010;Gálvez et al., 2016;Simpson et al., 2005), i.e., the snowpack can release gas-phase iodine. Firstly, this local iodine source needs to be added in the "origins and cycling of iodine" section in the introduction. Secondly, the manuscript is lacking discussion on photochemical post-depositional processes related to iodine loss from the snowpack. Such processes need to be quantified, especially at individual ice core sites due to their relationship with snow accumulation, before archived concentrations of iodine can be interpreted. Do you have any surface snow and atmospheric iodine measurements from Renland you could use? At the very least, you could make some assumptions that post-depositional processes are negligible at relatively high accumulation sites, such as Renland, but this would need to be backed up by evidence from the literature.

Response:We do not have atmospheric iodine measurements at the coring site but iodine recycling in surface snow and ice in the Arctic have been fully discussed in recent field experiments in Svalbard where similar photochemical processes on snow and ice

can be found (Spolaor et al., 2019). According to this study (and previous laboratory (Galvez et al., 2016; Kim et al., 2016) and field studies (Frieb et al., 2000; Simpson et al., 2005; Spolaor et al., 2014) iodine concentrations from in polar regions may suffer from significant photolytical post-depositional processes that affect the iodine concentration in ice at daily to seasonal scales. Nevertheless, significant changes in the net iodine depositional fluxes due to loss of iodine from snow at centennial to millennial time scales in coastal Greenland ice cores are not expected. We have now added a section in the revised manuscript where all the possible post-depositional processes affecting iodine re-emission from surface snow at daily to millennial time-scales are fully addressed.

Referee #1:The authors have produced a nice schematic diagram (Figure 3) showing the three phases of iodine evolution over the Holocene. The figure is only mentioned once in text and I recommend a fuller explanation of the figure throughout the discussion.

Response:We have added more explanation in the revised version of the manuscript regarding the conceptual model shown in Fig. 3

Referee #1:The study uses a combination of ice core and modelling evidence. The manuscript is rather ice core heavy and I recommend extending the modelling results and discussion sections especially for ice core readers who are not experts in atmospheric chemistry.

Response:We have extended the 2.3 section (Atmospheric Chemistry modelling) including more detail about the models run and the purpose of the experiments. Besides, we have also extended the result section of the manuscript with more information about the model results.

Referee #1:Lastly, I suggest restructuring the manuscript to combine the results and discussion into one section, and making the final conclusions separate. That would leave the reader with the key take home messages of the study.

Response:We have restructured the manuscript combining results and discussion and adding a conclusion section as recommended by reviewer.

Technical corrections Referee #1:L17 Insert "the" between "large influence on...oxidising capacity"

Response:Done

Referee #1:L21 Spell out "ReCAP" acronym

Response:Done

Referee #1:L21 Which region does your atmospheric iodine represent?

Response:The North Atlantic, we have added this information to the abstract in the new version of the manuscript

Referee #1:L23 Delete "-" between "before...present"

Response:Done

Referee #1:L23 Replace "ocean" with "marine"

Response:Done

Referee #1:L24 "Biological iodine explosion" is this term reported previously or are you suggesting it? Needs a little further explanation.

Response:We are suggesting this term. We have elaborated on that concept in the discussion in the new version of the manuscript

Referee #1:L24 What is the "iodine trend" doing during the early Holocene?

Response:We have rephrased it with "the high and stable iodine levels"

Referee #1:L29 This sentence is out of place. The paragraph is about the impact of halogen chemistry on the oxidising capacity of the atmosphere and thus your topic sentence should reflect this.

Response:The sentence has been deleted in the new version of the manuscript

Referee #1:L30 Remind the reader what time period the Holocene encompasses.

Response:Done

Referee #1:L32 Please add reference.

Response:Done

Referee #1:L45 Throughout the manuscript, you refer to ozone as "ozone", "tropospheric ozone", "atmospheric ozone", "surface ozone". Please be consistent with your terminology.

Response:We have now referred as tropospheric throughout the manuscript

Referee #1:L51 Here you need a greater explanation of the recycling processes of iodine in surface snow.

Response:We have added a complete section regarding post-depositional processes in ice and snow in the new version of the manuscript

Referee #1:L58 Capital "Ice".

Response:Done

Referee #1:L60 You states there are only two iodine records from ice cores. However, later in the paragraph you mention other records. Please add references for all iodine ice core records (e.g. Spolaor et al., 2013; Cuevas et al., 2018; Legrand et al., 2018).

Response:We have rephrased it adding all the iodine records available up to date in polar regions

Referee #1:L66 What does the Talos Dome iodine record reflect?

Response:Talos dome iodine record highlighted that sea-ice dynamics, algal productivity and dust controlled iodine variability at millennial time-scales variability in the

Antarctic region (Spolaor et al., 2013). We have added this information in the new version of the manuscript.

Referee #1:L72 Add the reference for the "recent study" upfront.

Response:Done

Referee #1:L86 Add "the" between "plateau and...Scoresby"

Response:Done

Referee #1:L91 Please add reference for Re-CAP age model.

Response:We have added the reference of the ReCAP age-depth model (Simonsen et al., in press)

Referee #1:L101 Were samples transported frozen?

Response:The samples were melted for sampling and frozen immediately after sampling. They were then kept frozen and away from light sources until they were analysed.

Referee #1:L102 Replace "of" with "at"

Response:Done

Referee #1:L102 "Curtin University of Technology" is now "Curtin University". Please replace here and throughout the manuscript. See https://www.curtin.edu.au/

Response:Done

Referee #1:L105 Change "CUT" to "CU" here and throughout.

Response:Done

Referee #1:L105 Please add reference to the Curtin University ICP-MS method or if these are the first measurements for iodine reported from that lab then please add methods and a table in the supplement with the operating parameters (e.g. torch, spray chamber, nebuliser).

Response:The details on the Curtin University ICP-MS method is fully described in Maffezzoli et al., 2018, we have added that reference in the new version of the manuscript.

Referee #1:L106 You define the Italian ICP-MS as "IDPA-CNR" please make sure you are consistent with your terminology throughout the manuscript.

Response:Done

Referee #1:L107 Reference is repeated twice. Please delete one.

Response:Done

Referee #1:L107 Add resolution for sodium measurements.

Response:This information has been added in the section 2.2 of the revised manuscript

Referee #1:L109 Spell out ultrapure water "UPW" acronym.

Response:Done

Referee #1:L109 What are these instrumental errors? Are they related to reproducibility or accuracy? Please add accuracy and precision for iodine and sodium measurements. What certified reference material did you use?

Response:The error bars on each data point reflect the standard deviation (1sigma) of the set of 5 mass scans performed by the ICP-MS for each sample detection. Therefore, they reflect the single-measurement precision. As Standard Reference Material we used one of the standards (NIST traceable commercial standards: High-Purity Standards, (Charleston, USA)) used during the calibrations as Quality Controlled Standards (QCs, as described in Maffezzoli et al., 2018, Appendix A1), continuously monitored during the analyses, for a total of n=82 detections. The spread (relative standard deviation) of the n=82 QCs concentrations (precision) was: 16% (iodine); 4% (sodium). The accuracy (as the ratio between the measured QCs concentration and the nominal QCs concentration) was on average 95±29% (iodine); 98±7% (sodium), See plots

from supplementary figure A

Referee #1:L110-111 Use consistent terminology for Italian and Australian ICP-MS.

Response:Done

Referee #1:L113 Yes there is a strong correlation between the two labs but it is non-linear at low concentrations. Include a note on the large area at low iodine concentrations.

Response:We agree with reviewer on the non-linear relation at low concentrations and we have added a note in the text regarding this. Nevertheless, the HTM iodine concentrations would still be significantly higher than the Neoglacial and the late Holocene iodine values, even accounting for the concentration non-linearity.

Referee #1:L116 Add "our" between" model and...sampling".

Response:Done

Referee #1:L116 Add "mass" between "depositional...fluxes".

Response:Done

Referee #1:L118 Please add reference for accumulation rates.

Response:Done

Referee #1:L119 Please add the different climatic phases in the introduction.

Response:Done

Referee #1:L121 Calcium is not mentioned previously. Either include Ca methods or remove entirely.

Response:We have added Ca methods in the new version of the manuscript

Referee #1:L124MBL spell out acronym.
[Figure]

Response:Done

Referee #1:L128 First mention of photolytic processes. Need to include iodine snow photochemistry in the introduction.

Response:In line 128 (within the Atmospheric Chemistry modelling section) we are referring to the photochemistry in the gas phase, not to the photochemistry in the snowpack. In this work, we are using THAMO to estimate the amounts of inorganic and organic iodine emissions that would reproduce the tropospheric iodine levels recorded in the ice core. Therefore, we are referring to the photolysis of the different species emitted to the troposphere, and more specifically to the photolysis of the different reactive iodine precursors (inorganic: HOI, I2 and organic: CH3I, CH2I2, CH2IBr and CH2ICl). Nevertheless, we have included a note in the supplementary information concerning the post-depositional processes.

Referee #1:L132 HTM spell out acronym.

Response:We have spelled out the acronym in the introduction in the new version of the manuscript

Referee #1:L132Delete "rs" in ""ky-1".

Response:Done

Referee #1:L135 Replace "results" with "ice core and modelling results" and add a fuller explanation of the modelling results here. Could you add a figure or table summarising the results?

Response:We have modified this heading as "Results and Discussion" in the new version of the manuscript. We have added Table 1 into the main text and we have added a new figure in the supplementary material summarizing the THAMO modelling main results

Referee #1:L137-138 Symbol$\mu$.

Response:Corrected

Referee #1:L139 What causes re-mobilisation?

Response:There are several wide range of factors that may contribute to iodine re-mobilization from surface snow and ice (e.g. photo-activation of reactive iodine, ice/firn characteristics, effect of snowfall rates, changes in meteorological conditions, etc) summarized in the literature (Frieb et al., 2000; Spolaor, et al 2014; Galvez et al., 2016; Kim et al., 2016; Legrand et al., 2018, Cuevas et al., 2018; Spolaor et al., 2019). The different mechanisms controlling iodine re-mobilization have been fully explained in a full section in the revised version of the manuscript.

Referee #1:L140 Please add reference of iodine concentration and accumulation rate studies. You haven't quantified how the accumulation rate impacts iodine loss at the Reland site so please include some evidence to justify that your assumption that post-depositional processes are negligible.

Response:We have added key references to the different accumulation rate studies in Greenland (e.g. Maselli et al., 2017, Rhodes et al., 2017). As discussed above, we have added a new section in the revised version of the manuscript regarding post-depositional effects on ReCAP site.

Referee #1:L154 Symbol$\mu$.

Response:Corrected

Referee #1:L147 What are the biomarkers?

Response:Biomarkers are summarized in Table S2 and Fig. 1 and explained below in the text.

Referee #1:L166 "The increase in nutrient supply from terrestrial sediment delivery".

Response:This has been corrected in the new version of the manuscript.

Referee #1:L172 Replace "The"with "An".

Response:Done

Referee #1:L174 Can you provide a reference "biological iodine explosion".

Response:"Biological Iodine Explosion" is a term suggested in this study for the first time.

Referee #1:L184 ssa acronym.

Response:ssa acronym has been previously spelled out in line 57 of the former version of the manuscript.

Referee #1:L185 Please mention calcium dust proxy and add reference.

Response:We have mention calcium as a dust proxy in Greenland and provided key references (i.e. (Schüpbach et al., 2018))

Referee #1:L185-186 Is this result from your model?

Response:Yes, we have specified it in the text now

Referee #1:L193 Replace "on" with "of".

Response:We have not modified it since we consider " on" is more appropriate in this case

Referee #1:L186 Period.

Response:Ok, we have rephrased it accordingly

Referee #1:L194 Replace"value" with "level".

Response:Done

Referee #1:L195 Add location of Kara and Laptev Seas to Fig. 1.

Response:Their location cannot be shown in Fig. 1 since they are located in the

Russian Arctic. We have explained its location in the text in the new version of the manuscript

Referee #1:L200 What measure of sea ice referring to? Sea ice extent or concentration or thickness? Here and throughout manuscript please specify.

Response:Sea ice extent and thickness we have clarified this throughout the manuscript

Referee #1:L204 How much thicker does the sea ice need to be? Can you give estimates of thickness and light penetration depth?

Response:In Antarctica, the light transmitted through sea-ice (with an average depth of ∼50 cm) has significant transmission down to ∼1m (King et al., 2005). This contrast with the sea-ice thickness in the Arctic (∼3 m), where iodine emissions are not significant (Saiz-Lopez et al., 2015). Therefore, a sea-ice thickness of ∼3 m would lead to negligible emissions of iodine from sea ice. We have now mention this in the manuscript.

Referee #1:L204 Add sea ice as a source of iodine in the introduction.

Response:Sea-ice as a source of iodine is new mentioned in the introduction in the new version of the manuscript explaining the biogenic production of HOI and I- from algae underneath the sea ice and its subsequent diffusion through brine channels to the atmosphere.

Referee #1:L211 Should the fluxes and concentrations be reported as 2 significant figures?

Response:Iodine concentration and fluxes curves are shown in Fig. 2 and in Figs. S2 and S3 for further details. Both fluxes and concentrations are shown in the same figures to allow intercomparison between them.

Referee #1:L222 Can you quantify the frequency? L223 What is the second part?

Response:We have rephrased the sentence as follows in order to make it more readable "The iodine time series in this part of the record reveals a higher variability mainly due to a reduced ice compression towards the surface and thus an increased temporal resolution of the samples"

Referee #1:L247 What are the associated radiative impacts?

Response:In this line we are referring to the impacts on the radiative balance of the atmosphere and therefore on climate. These impacts are on tropospheric ozone (Hossaini et al., 2015; Saiz-Lopez et al., 2012; Sherwen et al., 2017) and particle formation (Allan et al.,2015; Roscoe et al., 2015, Sipilä et al., 2016), as already commented at the end of the first paragraph of the Introduction section. We have now included these impacts in the discussion and provided the related references.

Referee #1:Figure 1 Mention the references in the caption to help the reader.

Response:Done

Referee #1:Figure S1 What does the 1:1 line indicate? Be consistent with the naming of the Australian and Italian iodine measurements.

Response:The 1:1 line indicates the x=y relation (it is not a fit). We have modified the naming according to reviewer suggestion

References Allan, J. D., Williams, P. I., Najera, J., Whitehead, J. D., Flynn, M. J., Taylor, J. W., Liu, D., Darbyshire, E., Carpenter, L. J., Chance, R., Andrews, S. J., Hackenberg, S. C., and McFiggans, G.: Iodine observed in new particle formation events in the Arctic atmosphere during ACCACIA, Atmos. Chem. Phys., 15, 5599-5609, 2015

Cuevas, C. A., Maffezzoli, N., Corella, J. P., Spolaor, A., Vallelonga, P.,Kjær, H. A., Simonsen, M., Winstrup, M., Vinther, B., and Horvat, C.: Rapid increase in atmospheric iodine levels in the North Atlantic since the mid-20th century, Nature communications, 9, 1452, 2018. Frieß, U., Deutschmann, T., Gilfedder, B., Weller,R., and Platt, U.: Iodine monoxide in the Antarctic snowpack, Atmospheric Chemistry and Physics, 10,

2439-2456, 2010. Gálvez, Ó., Baeza-Romero, M. T., Sanz, M., and Saiz-Lopez, A.: Photolysis of frozen iodate salts as a source of active iodine in the polar environment, 2016. Hossaini, R., Chipperfield, M., Montzka, S., Rap, A., Dhomse, S.and Feng, W., Efficiency of short-lived halogens at influencing climate through depletion of stratospheric ozone, Nature Geoscience, 8(3), 186, 2015.

Kim, K., Yabushita, A., Okumura, M., Saiz-Lopez, A., Cuevas, C. A., Blaszczak-Boxe, C. S., Min, D., W. Yoon, H.-I. and Choi, W., Production of molecular iodine and tri-iodide in the frozen solution of iodide: implication for polar atmosphere, Environmental science & technology, 50(3), 1280-1287, 2016.

King, M. D., France, J. L., Fisher, F. N., & Beine, H. J.. Measurement and modelling of UV radiation penetration and photolysis rates of nitrate and hydrogen peroxide in Antarctic sea ice: An estimate of the production rate of hydroxyl radicals in first-year sea ice. Journal of photochemistry and photobiology A: Chemistry, 176(1-3), 39-49. 2005

Legrand, M., McConnell, J. R., Preunkert, S., Arienzo, M.,Chellman, N., Gleason, K., Sherwen, T., Evans, M. J., and Carpenter, L. J.: Alpine ice evidence of a three-fold increase in atmospheric iodine deposition since 1950 in Europe due to increasing oceanic emissions, Proceedings of the National Academy of Sciences, 115, 12136-12141, 2018. Maffezzoli, N., Vallelonga, P., Edwards, R., Saiz-Lopez, A., Turetta, C., Kjær, H. A., Barbante, C., Vinther, B. and Spolaor, A., 120,000 year record of sea ice in the North Atlantic, Climate of the Past Discussions, https://doi.org/10.5194/cp-2018-80, 2018.

Maselli, Olivia J., Chellman, Nathan J., Grieman, Mackenzie, Layman, Lawrence, McConnell, Joseph R., Pasteris, Daniel, Rhodes, Rachael H., Saltzman, Eric and Sigl,Michael. Sea ice and pollution-modulated changes in Greenland ice core methane sulfonate and bromine. Climate of the Past, 13 (1). pp. 39-59. 2017

Rhodes, R. H., Yang, X., Wolff, E. W., McConnell, J. R., & Frey, M. M. Sea ice as a

source of sea salt aerosol to Greenland ice cores: a model-based study. Atmospheric Chemistry and Physics, 17(15), 9417-9433. 2017 Roscoe, H. K., Jones, A. E., Brough, N., Weller, R., Saizâ ĂŘLopez, A., Mahajan, A. S., Schoenhardt, A., Burrows, J. P., and Fleming, Z. L., Particles and iodine compounds in coastal Antarctica, Journal of Geophysical Research: Atmospheres, 120(14), 7144-7156, 2015 Saiz-Lopez, A., Plane, J. M., Baker, A. R., Carpenter, L. J., von Glasow, R., GoÌ Ąmez MartiÌ Ąn, J. C., McFiggans, G. and Saunders, R. W., Atmospheric chemistry of iodine, Chemical reviews, 112(3), 1773-1804, 2012.

Saiz-Lopez, A., Blaszczak-Boxe, C.S. and Carpenter, L., A mechanism for biologically induced iodine emissions from sea ice, Atmospheric Chemistry and Physics, 15(17), 9731-9746, 2015. Schüpbach, S., Fischer, H., Bigler, M., Erhardt, T., Gfeller, G., Leuenberger, D.et al. Greenland records of aerosol source and atmospheric lifetime changes from the Eemian to the Holocene. Nature communications, 9(1), 1476. 2018

Sherwen, T., Evans, M. J., Carpenter, L. J., Schmidt, J. A., and Mickley, L. J., Halogen chemistry reduces tropospheric O3 radiative forcing, Atmos. Chem. Phys., 17, 1557-1569, 2017. Sipilä, M., Sarnela, N., Jokinen, T., Henschel, H., Junninen, H., Kontkanen, J., Richters, S., Kangasluoma, J., Franchin, A. and Peräkylä, O., Molecular-scale evidence of aerosol particle formation via sequential addition of HIO3, Nature, 537(7621), 532, 2016 Spolaor, A., Vallelonga, P., Plane, J., Kehrwald,N., Gabrieli, J., Varin, C., Turetta, C., Cozzi, G., Kumar, R., and Boutron, C.: Halogenspecies record Antarctic sea ice extent over glacial–interglacial periods, Atmospheric Chemistry and Physics, 13, 6623-6635, 2013. Simonsen, M.F., Baccolo, G., Blunier, T., Borunda, A., Delmonte, B., Frei, R., Goldstein, S., Grinsted, A., Kjær, H.A., Sowers, T., Svensson, A., Vinther, B., Vladimirova, D., Winckler, G., Winstrup, M. and Vallelonga, P.. East Greenland ice core dust record reveals timing of Greenland ice sheet advance and retreat. Nature Communications, 2019, in press.

Spolaor, A., Barbaro, E., Cappelletti, D., Turetta, C., Mazzola, M., Giardi, F., Björkman, M.P., Lucchetta, F., Dallo, F., Pfaffhuber, K.A., Angot, H., Dommergue, A., Maturilli, M.,

Saiz-Lopez, A., Barbante, C. and Cairns, W.R.L. Diurnal cycle of iodine and mercury concentrations in Svalbard surface snow. Atmos. Chem. Phys. Discuss. 2019, 1-25. 2019

[Figure]

[Figure]

**Fig. 1.** Supplementary figure A: Iodine and sodium accuracy in the measured samples

---

## Author Comment (AC2) · 26 Sep 2019

Referee #3: In the paper "Holocene atmospheric iodine evolution over the North Atlantic" by J. P.Corella and co-authors a new record of iodine concentration in the ReCAP ice core (Greenland) is presented. The record covers the full Holocene and an interesting discussion about iodine variations in the atmosphere across different climatic periods is shown. In particular the authors elaborate here an easy conceptual model which seems to be very effective in explaining iodine variability during the three time periods here considered: Holocene Thermal Maximum, Neoglacial period and Great Acceleration. The paper shows a very good laboratory intercomparison exercise, showing a general agreement in the measured iodine concentrations. This work is generally well presented and well written and will surely be a milestone in the use

of halogens measured in ice cores to achieve information about Sea Ice in the North Atlantic.

Response: We appreciate the positive feedback from reviewer 3. We have addressed below all the comments and suggestions made by reviewer.

Referee #3: My main concern about the paper is the lack of discussion about the possible post-depositional effects of iodine species in the snow. A detailed discussion about the preservation of iodine compounds in the snow and their stability in time due to a possible photochemical degradation should be added to the text. Before concluding that iodine is a real marker reflecting the past North Atlantic conditions, the authors should assess that the very low concentrations they are measuring are really telling us a story about past conditions, or, at least, that the record obtained is not significantly different from what was deposited in the Renland Ice Cap.

Response: We agree with the reviewer on the fact that photochemical processes on snow and ice lead to iodine recycling for the surface snow. Indeed, several laboratory and field experiments reported in the literature (e.g Frieb et al., 2000; Spolaor et al. 2013; Galvez et al., 2016; Kim et al., 2016; Spolaor et al., 2019) have shown that iodine concentrations from in polar regions may suffer from significant photo-chemical post-depositional processes that may affect the iodine concentration in ice and snow at daily to seasonal scales. Nevertheless, large variations in the net iodine depositional fluxes in coastal Greenland ice cores at centennial to millennial time scales are not expected. We have added a section in the new version of the manuscript where the post-depositional processes affecting iodine re-emission from ice and snow at different time scales are fully addressed

Referee #3: Moreover, the meteorological conditions were surely very different in the different time periods here shown, and the authors should clarify why the variability of iodine is mainly linked to the primary productivity and not to changes in transport efficiency through time.

Response: It is difficult to reconstruct meteorological conditions accurately in Greenland during the Holocene, but the best indicator we have of the conditions at the site is the reconstructed accumulation rate shown in Figure S2. This was reasonably stable for the last 8000 years, suggesting relatively constant snowfall and surface conditions. The absence of any significant correlation between the Holocene accumulation rate and the concentration of impurities such as iodine and ssa, is the first indication that the observed pattern of those impurities is not dominated by snow deposition or meteorological effects. The scale of variability exhibited in the RECAP Iodine record (both concentrations and fluxes) is further inconsistent with possible changes in transport efficiency. The iodine concentration changes by more than a factor of 4 within a few centuries during the Holocene, which is an enormous change. For comparison, best-available estimates suggest only a minor change in transport efficiency (much less than a factor of 4) over the glacial-interglacial transition for dust travelling to Greenland from deserts in central Asia (Schüpbach et al., 2018)

Referee #3: Other papers by the authors reported an halogen enrichment factor by subtracting the sea salt contribution. I'm not sure about which is the best way to proceed, but I would like to know why in this paper this elaboration was not considered. I suspect that an elaboration of the iodine values with sodium can infer to iodine a variability which mainly comes from Na+ concentrations. Once these points will be addressed I would recommend the publication of this paper.

Response: As suggested by reviewer, we have added a new figure (Fig. S5) showing the iodine to sodium ratio. In this figure we show that the I/Na ratio show the same trends that the iodine concentration and fluxes throughout the Holocene reinforcing our interpretation that ssa would only represent a minor contributor explaining the iodine variability in ReCAP during the Holocene. We have added this new information in the revised manuscript.

Minor issues.

Referee #3: Line 59: Change to "Ice core..."

Response: Done

Referee #3: Line 109: explain what UPW means.

Response: Done

Referee #3: Line 110: How were the d.l. calculated? Explain briefly how the two labs measured this figure of merit of the two methods.

Response: This information has been added to the methods section in the new version of the manuscript

Referee #3: Line 116: Is there an age model for the RECAP ice core? Give more information on this point: how was the age scale built? Which absolute markers were used (i.e. volcanoes, tephra, etc)? Which is the uncertainty in the bottom part of the ice core?

Response: The ReCAP ice core record covers the last 120 kyr BP and has been published in Simonsen et al., 2019. All the details regarding the chronology are fully explained. Addressing the specific questions made by reviewer: -The RECAP timescale down to 458.3 m (4048.1 y b2k) was produced using the StratiCounter automated layer counting software (https://github.com/maiwinstrup/StratiCounter) (Winstrup, 2016). -The software was constrained to fit 28 volcanic marker horizons dated in other Greenland ice cores (DYE-3, GRIP, GRIP and NEEMS1) formalised in the Greenland Ice Core Chronology 2005 [Rasmussen et al., 2013]. Below 458.3 m, the timescale was derived automatically using a shape-preserving piecewise cubic interpolation (Vinther, et al., 2008) between 15 chronological tie points (volcanic markers or climate transition matches) used to constrain the timescale back to 11703 y b2k. -As the RECAP timescale is synchronised to GICC05, it inherits the uncertainty budget associated with that timescale and presented in the aforementioned references.

Referee #3: Line 137-138: units are not readable...

Response: This has been corrected in the new version of the manuscript

Referee #3: Line 141: remove the comma between m-2 and yr-1.

Response: Done

Referee #3: Line 198: Two references by Telesinski et al. were published in 2014, please call them a) and b) in order to make clear which reference you are referring to in the text.

Response: Done

Referee #3: Line 198: is c. for ca.?

Response: YES, this error has been corrected.

Referee #3: Line 240: remove M.M. before Telesinski.

Response: Done

Referee #3: Line 412: check the references: something wrong happened to the title of this paper.

Response: The typos from the title reference title have been corrected now

Referee #3: Figure 2 caption: use "concentration and flux" or "concentrations and fluxes", not a mix

Response: Done

Referee #3: Figure 2: during the HTM the authors' reconstruction seems to be very smoothed with respect to the marine records. Is it due to the poor resolution at this depth? Please discuss this point in the text.

Response: Yes, this is due to the lower time resolution (same 55 cm sampling resolution but increased ice compression) at this depth. This has been discussed in the new version of the manuscript. "The iodine and sodium measurement temporal resolution

ranged from sub-annual in the upper metres during the Great Acceleration (average resolution of 1.31 samples/yr), subdecadal resolution during the Late Holocene (average resolution of 0.23 samples/yr), and sub-centennial resolution during the Neoglacial and the HTM (average resolution of 0.035 and 0.011 samples/yr respectively) according to the RECAP ice core age-depth model and our sampling resolution".

Referee #3: Figure 2 and text: is flux adding something new with respect to concentration? The two profiles are very similar and the authors should discuss this point in terms of wet/dry deposition mechanisms at the drilling site.

Response: According to Wolff et al 2006 if the accumulation rate is low, dry deposition dominates and fluxes are proportional to the air concentration, while if the accumulation rate is high, wet deposition dominates the dry deposition, the concentration is proportional to the air concentration. Therefore, since accumulation rates in ReCAP are much higher than other Greenland sites summarized in Maselli et al (2017) and Rhodes et al (2017) we may suggest that wet deposition dominates. Nevertheless, this relation is not straight forward and we prefer to add both concentration and fluxes. Both iodine concentration and fluxes shown similar trends. This agreement between concentration and fluxes reinforces our explanation of external forcings (i.e. marine bioproductivity and sea ice dynamics, driving the iodine levels variability throughout the Holocene. We have mentioned this in the revised version of the manuscript.

References Frieß, U., Deutschmann, T., Gilfedder, B., Weller,R., and Platt, U.: Iodine monoxide in the Antarctic snowpack, Atmospheric Chemistry and Physics, 10, 2439-2456, 2010 Gálvez, Ó., Baeza-Romero, M. T., Sanz, M., and Saiz-Lopez, A.: Photolysis of frozen iodate salts as a source of active iodine in the polar environment, 2016. Kim, K., Yabushita, A., Okumura, M., Saiz-Lopez, A., Cuevas, C. A., Blaszczak-Boxe, C. S., Min, D., W. Yoon, H.-I. and Choi, W., Production of molecular iodine and tri-iodide in the frozen solution of iodide: implication for polar atmosphere, Environmental science & technology, 50(3), 1280-1287, 2016.

Maselli, Olivia J., Chellman, Nathan J., Grieman, Mackenzie, Layman, Lawrence, McConnell, Joseph R., Pasteris, Daniel, Rhodes, Rachael H., Saltzman, Eric and Sigl,Michael. Sea ice and pollution-modulated changes in Greenland ice core methane sulfonate and bromine. Climate of the Past, 13 (1). pp. 39-59. 2017

Rhodes, R. H., Yang, X., Wolff, E. W., McConnell, J. R., & Frey, M. M. Sea ice as a source of sea salt aerosol to Greenland ice cores: a model-based study. Atmospheric Chemistry and Physics, 17(15), 9417-9433. 2017 Simonsen, M.F., Baccolo, G., Blunier, T., Borunda, A., Delmonte, B., Frei, R., Goldstein, S., Grinsted, A., Kjær, H.A., Sowers, T., Svensson, A., Vinther, B., Vladimirova, D., Winckler, G., Spolaor, A., Vallelonga, P., Plane, J., Kehrwald,N., Gabrieli, J., Varin, C., Turetta, C., Cozzi, G., Kumar, R., and Boutron, C.: Halogenspecies record Antarctic sea ice extent over glacial–interglacial periods, Atmospheric Chemistry and Physics, 13, 6623-6635, 2013.

Spolaor, A., Barbaro, E., Cappelletti, D., Turetta, C., Mazzola, M., Giardi, F., Björkman, M.P., Lucchetta, F., Dallo, F., Pfaffhuber, K.A., Angot, H., Dommergue, A., Maturilli, M., Saiz-Lopez, A., Barbante, C. and Cairns, W.R.L. Diurnal cycle of iodine and mercury concentrations in Svalbard surface snow. Atmos. Chem. Phys. Discuss. 2019, 1-25. 2019

Schüpbach, S., Fischer, H., Bigler, M., Erhardt, T., Gfeller, G., Leuenberger, D.et al. Greenland records of aerosol source and atmospheric lifetime changes from the Eemian to the Holocene. Nature communications, 9(1), 1476. 2018

Winstrup, M. and Vallelonga, P.. East Greenland ice core dust record reveals timing of Greenland ice sheet advance and retreat. Nature Communications, 2019, in press.

Vinther, B. M. et al. Synchronizing ice cores from the Renland and Agassiz ice caps to the Greenland Ice Core Chronology,. J. Geophys. Res. 113, D08115, doi:08110.01029/02007JD009143. 2008

Winstrup, M. A Hidden Markov Model Approach to Infer Timescales for High-Resolution

Climate Archives. (AAAI Press). 2016

Wolff, E. W., Fischer, H., Fundel, F., Ruth, U., Twarloh, B., Littot, G. C., ... et al. Southern Ocean sea-ice extent, productivity and iron flux over the past eight glacial cycles. Nature, 440(7083), 491. 2006

[Figure]

[Figure]

**Fig. 1.** Fig. S4: ReCAP ice core iodine to sodium concentration ratio during the Holocene

---

## Author Comment (AC3) · 26 Sep 2019

Referee #2: In this study, the first ice core iodine record spanning the Holocene is presented. Iodine levels were high during the early Holocene and industrialization relative to the late Holocene. Chemical transport model results of inorganic iodine sources and transport are compared to ice core iodine levels. This comparison suggests that marine biogenic sources of iodine were higher during the early Holocene relative to the late Holocene. This increase in biogenic sources is reasonable in the early Holocene given that other reconstructions show that sea surface temperatures in the North Atlantic were warmer, salinity was higher, sea ice extent was lower, and primary productivity of subpolar planktonic species was higher. This study is well-written and well-organized. It should be published because it is the first record iodine covering the early Holocene

and provides strong links to climatic factors that could explain the unexpectedly high levels of early Holocene iodine. Hopefully, it will inspire future study of iodine species in ice cores.

Response: We appreciate the positive feedback from reviewer 2. Please, find below the responses to all the comments made by referee #2

Major comments

Referee #2: Section 1, lines 36-37: This sentence is confusing. What exactly is meant by "has a global contribution of up to 27% of the total rate of ozone loss?"

Response: We are referring to reactive iodine as an important ozone-depleting family in the troposphere. Specifically, we are saying that reactive iodine is responsible of up to 27% of the ozone loss in both the marine boundary layer and the upper troposphere. We have rephrased that statement to clarify the meaning.

Referee #2: Section 1, lines 58-68: Two ice cores are reported as having the only iodine records, but further in the paragraph, a third ice core record is discussed. Please clarify.

Response: We have modified the sentence to clarify the iodine reconstruction in polar regions available up to date

Referee #2: Section 2.1, line 91-96. Please discuss the depth-age scale in more detail because (it is not yet published?). Perhaps include a depth-age scale in the supplement. Which volcanic markers were used and how well-spaced are they throughout the record? What are the uncertainties?

Response: The ReCAP ice core record covers the last 120 kyr BP and is currently in press (Simonsen et al., 2019). The RECAP timescale down to 458.3 m (4048.1 y b2k) was produced using the StratiCounter automated layer counting software (https://github.com/maiwinstrup/StratiCounter) [Winstrup, 2016)]. The software was constrained to fit 28 volcanic marker horizons dated in other Greenland ice cores

(DYE-3, GRIP, GRIP and NEEMS1) formalized in the Greenland Ice Core Chronology 2005 (Rasmussen et al., 2013). Below 458.3 m, the timescale was derived automatically using a shape-preserving piecewise cubic interpolation (Vinther, et al., 2008) between 15 chronological tie points (volcanic markers or climate transition matches) used to constrain the timescale back to 11703 y b2k. As the RECAP timescale is synchronized to GICC05, it inherits the uncertainty budget associated with that timescale and presented in the aforementioned references.

Referee #2: Section 2.2, lines 99-100. Please include the sampling resolution at various depths. Clearly, several more samples were analysed in more recent years which makes sense as the fractions were collected via the CFA system. The easiest way to do this may be to include a mean sampling resolution for the time periods discussed in each section (results).

Response: The iodine and sodium measurement temporal resolution ranged from sub-annual in the upper metres during the Great Acceleration (average resolution of 1.31 samples/yr), subdecadal resolution during the Late Holocene (average resolution of 0.23 samples/yr), and sub-centennial resolution during the Neoglacial and the HTM (average resolution of 0.035 and 0.011 samples/yr respectively). This information has been added in the revised manuscript

Referee #2: Section 3: Please use the same units when possible. It is difficult for the reader to understand readily the comparisons between different periods of time and the model versus the observations when the units are changed.

Response: We have now changed the unit of the modelling results throughout the whole text from nmol m-2 d-1 to $\mu$g m-2 y-1 according to the observations units.

Referee #2: Section 3: It would be useful to discuss the relationship between concentration and flux here. Did the flux calculation confirm that remobilization processes were not consequential? Are there any important differences when flux is used?

Response: In ReCAP ice core, reconstructed accumulation rates shown in Figure S2 shows reasonably stable levels for the last 8000 years, suggesting relatively constant snowfall and surface conditions. Variability in the snow accumulation (derived from annual layer thickness calculations) occurring prior to 8000 years ago may be due to glaciological flow effects already identified in the previous Renland ice core drilled in 1988 (Hansson et al., 1994) just 2 km from the site of the 2015 RECAP ice core (this work).The absence of any significant correlation between the accumulation rates and the concentration of impurities such as iodine and ssa also indicate that the Holocene iodine variability in ReCAP ice core is neither related to the effects of changing snow deposition rates. Furthermore, both iodine concentration and fluxes shown similar trends. This agreement between concentration and fluxes reinforces the explanation that source effects (i.e. marine bioproductivity and sea ice dynamics) rather than relative changes in wet/dry deposition are driving the iodine levels variability throughout the Holocene. We have mentioned this in the revised version of the manuscript.

Referee #2: Line 141: Start a new paragraph here? How does the second part of this paragraph relate to the first part?

Response: Done, we have modified this section by adding a new sub-section regarding re-mobilization processes in snow and ice

Referee #2: 3.1, lines 182-183 the authors state that there is no correlation. Is it possible to also include a time series of calcium and sodium measurements in the supplement?

Response: Correlation coefficients of sodium and calcium are shown in Table S3 and in section. Sodium profiles are shown in Maffezzoli et al (2018) while calcium dataset is available upon request. We have stated this in the revised manuscript

Referee #2: Lines 191: Please use the same units throughout the manuscript when possible.

Response: Done

Referee #2: Lines 192-193: How would the ssa contribution change over time? Is the lack of a correlation during the early Holocene purely due to the biogenic contribution overwhelming the other signals? Are they still there?

Response: Sea salt aerosol (ssa) inputs to RECAP are a combination of sea salt directly lofted from open ocean (wave breaking and storms) as well as sea salt incorporated into blowing snow over sea ice. When iodine is primarily emitted through organic processes there should be less correlation between ssa and iodine. When iodine is primarily emitted through inorganic processes, there should be a greater correlation. The reviewer's description is therefore accurate, that the lack of correlation during the early Holocene supports the interpretation of a dominance of organic iodine emission sources. As described in Supplementary Table S1, the THAMO model results indicate organic iodine emissions were effectively zero during the Neoglacial, but were active before and after the Neoglacial and continue to the present. Ssa showed an abrupt increased at the beginning of the Holocene and fluctuating values throughout the Holocene. This variability is described in Maffezzoli et al. (2018). During the HTM the biogenic contribution would be much higher than the iodine related to atmospheric ssa deposition. This is well reflected in the I/Na ratio that have been added to Fig. S4 in the revised version of the manuscript. However, ssa, as well as dust, could also be minor contributors of the iodine levels during the HTM. This has been reflected in the revised manuscript.

Referee #2: Lines 205-208: Please include the ice core values here and clarify how this conclusion was made, given the values provided.

Response: We have added the mean iodine depositional fluxes from the ice core during this period. The modelling results during this period show that inorganic iodine emissions were significantly higher than the organic iodine emissions (Table 1) which is in agreement with the decreased in bioproductivity shown in Fig. 2 and described

in the text. We have now provided the values for organic iodine emissions in the new version of Table S1 in order to make this clearer

Referee #2: Lines 221-223: Replace "a higher frequency" with "higher frequency variability." I agree that this is pretty clearly due to increased sampling resolution. If anything, averaging or smoothing could be used to compare the late Holocene to the early Holocene. The late Holocene levels may only appear to show higher frequency variability because there are more samples.

Response: We agree with reviewer. Thus, we have rephrased the sentence as follows: "The iodine time series in this part of the record reveals a higher frequency variability mainly due to a reduced ice compression towards the surface and thus an increased temporal resolution of the samples"

Referee #2: The focus of this study is iodine and should remain iodine. Do other halogens, like bromine, that were measured in the ReCAP ice core (Maffezzoli et al., 2018, in re-view), also show the same high levels during the early Holocene? Are the processes governing the sources, transport, and deposition of other halogens so different that they should not be included in the supplement?

Response: Neither Br nor Cl show that abrupt difference between Early Holocene and the rest of the Holocene. Bromine enrichments found in ReCAP ice core are mostly influenced by the autocatalytic processes in Multi-Year Vs First-Year sea-ice that are fully described in Maffezzoli et al (2018). On the other hand, Cl is dependent on ssa variability and other complex recycling processes in the polar atmosphere that are beyond the scope of this paper. Therefore, in our opinion it would not make sense to add the other halogens in the supplementary material.

Referee #2: Consider adding a section about the modelling results. These results are used to explain the interpretation of the ice core record, but it would be useful to have an understanding of the results prior to the comparison.

[Figure]

Response: We have elaborated on the modelling section. We have added more information about THAMO and described with more detail the outcome of the modelling results in the revised version of the manuscript. On top of that we have added a new figure in the supplementary material (Figure S3) that displays the modelled atmospheric reactive iodine according to the different oceanic iodine emissions during the present day and the different Holocene periods discussed in the text.

Figures and tables

Referee #2: fig. 1 add mapping software reference and color bar if going to include bathymetry

Response: We have added the software reference (Ocean Data View, Schlitzer, R., 2015, http://odv.awi.de) and the colour bar in the new version of the figure

Referee #2: fig.2 add indication of detection limits to iodine concentrations

Response: We have added the information of the detection limits in the Figure caption

Referee #2: Table S1: Please consider moving table S1 to the main text.

Response: Done, Table S1 has been incorporated to the main text as table 1. The table now includes the inorganic and organic oceanic emission in the same units of the ice core fluxes ($\mu$g m-2 y-1).

Referee #2: Please add more to this caption. What are the sources (ice core versus model inputs/outputs) in each column?

Response: We have now including more information in the caption of the table, defining the different fluxes included in the organic and inorganic iodine columns. Minor Comments:

Referee #2: Abstract, line 22: change to: "in the record were found"

Response: Done

Referee #2: line 30: "allows for the detailed"

Response: Done

Referee #2: line 32: "key factor for understanding"

Response: Done

Referee #2: line 42: change to "involve"

Response: Done

Referee #2: line 59: "Ice core"

Response: Done

Referee #2: line 66: "allow for a"

Response: Done

Referee #2: Rename section 2.1 "Study site and dating" or "Study site and age scale"

Response: Done

Referee #2: Line 108: "with the stability of the instrumental"

Response: Done

Referee #2: Line 126: remove "The model"

Response: Done

Referee #2: Line 129: "all of the"

Response: Done

Referee #2: Line 138: add comma before "respectively"

Response: Done

Referee #2: Line 141: remove "thus" before "Holocene"

Response: Done

Referee #2: Line 146: remove "occurring" before "before"

Response: Done

Referee #2: Line 203: change to "algal"

Response: Done

Referee #2: Line 205: change to "both of these processes"

Response: Done

Referee #2: Line207, 220: "concentrations"

Response: Done

Referee #2: Line 221: commas before and after "respectively"

Response: Done

Referee #2: Line 241:comma before "resulting"

Response:

Referee #2: Line 251: "to accelerate considerably"

Response: Done

References Hansson, M. E., The Renland ice core. A Northern Hemisphere record of aerosol composition over 120,000 years. Tellus B: Chemical and Physical Meteorology, 46(5), 390-418, 1994.

Legrand, M., McConnell, J. R., Preunkert, S., Arienzo, M.,Chellman, N., Gleason, K., Sherwen, T., Evans, M. J., and Carpenter, L. J.: Alpine ice evidence of a three-fold increase in atmospheric iodine deposition since 1950 in Europe due to increasing oceanic emissions, Proceedings of the National Academy of Sciences, 115, 12136-

12141, 2018

Maffezzoli, N., Vallelonga, P., Edwards, R., Saiz-Lopez, A., Turetta, C., Kjær, H. A., Barbante, C., Vinther, B. and Spolaor, A., 120,000 year record of sea ice in the North Atlantic, Climate of the Past Discussions, https://doi.org/10.5194/cp-2018-80, 2018.

Rasmussen, S. O. et al. A first chronology for the North Greenland Eemian Ice Drilling (NEEM) ice core. Clim. Past 9, 2713-2730. 2013. Schlitzer, R., 2015, Ocean Data View, http://odv.awi.de Simonsen, M.F., Baccolo, G., Blunier, T., Borunda, A., Delmonte, B., Frei, R., Goldstein, S., Grinsted, A., Kjær, H.A., Sowers, T., Svensson, A., Vinther, B., Vladimirova, D., Winckler, G., Winstrup, M. and Vallelonga, P.. East Greenland ice core dust record reveals timing of Greenland ice sheet advance and retreat. Nature Communications, 2019, in press.

Vinther, B. M. et al. Synchronizing ice cores from the Renland and Agassiz ice caps to the Greenland Ice Core Chronology,. J. Geophys. Res. 113, D08115, doi:08110.01029/02007JD009143. 2008

Winstrup, M. A Hidden Markov Model Approach to Infer Timescales for High-Resolution Climate Archives. (AAAI Press). 2016
* * *
**Fig. 1.** Figure S3. Diurnal variation of reactive iodine modelled by THAMO in different scenarios of [O3] and VSLs e missions fluxes

[Figure]

**Fig. 2.** Fig. 1: Location of the ReCAP ice core (red) and other marine paleoceanographic archives in the Nordic Seas discussed in the text.

---

## Author Response (AR1)

**Referee #1**

General comments Corella et al. present a relatively high-resolution ice core record of atmospheric iodine variability during the Holocene from the Renland Ice Core, Greenland. Iodine concentrations are extremely low in ice cores (<0.1 ng g-1), and require meticulous sample preparation and ultra-trace analytical protocols. This study presents a very nice lab intercalibration experiment between two ice core labs - Curtin University, Australia and IDPA-CNR, Italy, which I commend. Over the Holocene, the authors suggest the large variability in iodine concentrations and iodine fluxes, measured in the ice core, reflect changes in marine primary production over the North Atlantic. They derive a conceptual model to explain the three-step changes in iodine levels over the last 11 ka, i.e., Holocene Thermal Maximum, Neoglacial Period, and Late Holocene. This iodine record will be of interest to a number of communities including atmospheric chemists, palaeoclimatologists and marine biogeochemists. However, the authors have not addressed post-depositional processes that are known to modify the original iodine deposition to snow. These photochemical processes need to be quantified before ice core records of iodine can be interpreted at the Renland site. After addressing the issue of post-depositional processes, I am happy to recommend this manuscript for publication in Climate of the Past.

We appreciate the positive feedback from reviewer 1. We have addressed below all the comments and suggestions made by reviewer that have certainly improved the quality of the manuscript. In particular, we have added a detailed section (i.e. supplementary information) in which we evaluate the different post-depositional processes that may occur at the Renland site.

Specific comments

Post-depositional processes, such as UV-photolysis, can cause re-active halogen species in the snowpack to be lost to the overlying atmosphere (e.g.Frieß et al., 2010;Gálvez et al., 2016;Simpson et al., 2005), i.e., the snowpack can release gas-phase iodine. Firstly, this local iodine source needs to be added in the "origins and cycling of iodine" section in the introduction. Secondly, the manuscript is lacking discussion on photochemical post-depositional processes related to iodine loss from the snowpack. Such processes need to be quantified, especially at individual ice core sites due to their relationship with snow accumulation, before archived concentrations of iodine can be interpreted. Do you have any surface snow and atmospheric iodine measurements from Renland you could use? At the very least, you could make some assumptions that post-depositional processes are negligible at relatively high accumulation sites, such as Renland, but this would need to be backed up by evidence from the literature.

We do not have atmospheric iodine measurements at the coring site but iodine recycling in surface snow and ice in the Arctic have been fully discussed in recent field experiments in Svalbard where similar photochemical processes on snow and ice can be found (*Spolaor et al., 2019*). According to this study and previous laboratory (*Galvez et al., 2016; Kim et al., 2016*) and field studies (*Frieb et al., 2000; Simpson et al., 2005; Spolaor et al., 2014*) iodine concentrations in polar regions may suffer from significant photolytical post-depositional processes affecting iodine concentration in ice at daily to seasonal scales. Nevertheless, significant changes in the

net iodine depositional fluxes due to loss of iodine from snow at centennial to millennial time scales in coastal Greenland ice cores are not expected. We have now added a section in the SI of the revised manuscript where we have addressed all the possible post-depositional processes affecting iodine re-emission from surface snow at daily to millennial time-scales.

The authors have produced a nice schematic diagram (Figure 3) showing the three phases of iodine evolution over the Holocene. The figure is only mentioned once in text and I recommend a fuller explanation of the figure throughout the discussion.

We have referred more times to the conceptual model shown in Fig. 3 throughout the revised version of the manuscript

The study uses a combination of ice core and modelling evidence. The manuscript is rather ice core heavy and I recommend extending the modelling results and discussion sections especially for ice core readers who are not experts in atmospheric chemistry.

We have extended the 2.3 section (Atmospheric Chemistry modelling) including more detail about the models run and the purpose of the experiments (Pag 5, lines 132-153). Besides, we have also extended the result section of the manuscript with more information about the model results (Pag 7, lines 204-209; Pag 8, lines 235-239). We have also added a new figure in the supplementary material (Fig. S3) showing the diurnal variation of reactive iodine modelled by THAMO in different Holocene scenarios discussed in the text

Lastly, I suggest restructuring the manuscript to combine the results and discussion into one section, and making the final conclusions separate. That would leave the reader with the key take home messages of the study.

We have restructured the manuscript combining results and discussion and adding a conclusion section as recommended by reviewer.

Technical corrections

L17 Insert "the" between "large influence on...oxidising capacity"

Done

L21 Spell out "ReCAP" acronym

Done

L21 Which region does your atmospheric iodine represent?

The North Atlantic, we have added this information to the abstract in the new version of the manuscript (Pag 1, lines 21-22).

L23 Delete "-" between "before...present"

Done

L23 Replace "ocean" with "marine"

Done

L24 "Biological iodine explosion" is this term reported previously or are you suggesting it? Needs a little further explanation.

We are suggesting this term. We have elaborated on that concept in the discussion in the new version of the manuscript (section 3.1)

L24 What is the "iodine trend" doing during the early Holocene?

We have rephrased it with "the high and stable iodine levels" (Pag 1, line 25)

L29 This sentence is out of place. The paragraph is about the impact of halogen chemistry on the oxidising capacity of the atmosphere and thus your topic sentence should reflect this.

The sentence has been deleted in the new version of the manuscript

L30 Remind the reader what time period the Holocene encompasses.

Done

L32 Please add reference.

Done

L45 Throughout the manuscript, you refer to ozone as "ozone", "tropospheric ozone", "atmospheric ozone", "surface ozone". Please be consistent with your terminology.

We have now referred as tropospheric throughout the manuscript

L51 Here you need a greater explanation of the recycling processes of iodine in surface snow.

We have added a complete section regarding post-depositional processes in ice and snow in the new version of the manuscript (SI)

L58 Capital "Ice".

Done

L60 You states there are only two iodine records from ice cores. However, later in the paragraph you mention other records. Please add references for all iodine ice core records (e.g. Spolaor et al., 2013; Cuevas et al., 2018; Legrand et al., 2018).

We have rephrased it adding all the iodine records available up to date in polar regions (Pag2, lines 59-61).

L66 What does the Talos Dome iodine record reflect?

Talos dome iodine record highlighted that sea-ice dynamics, algal productivity and dust controlled iodine variability at millennial time-scales variability in the Antarctic region (*Spolaor et al., 2013*). We have added this information in the new version of the manuscript (Pags 2 and 3, lines 65-71).

L72 Add the reference for the "recent study" upfront.

Done

L86 Add "the" between "plateau and...Scoresby"

Done

L91 Please add reference for Re-CAP age model.

We have added the reference of the ReCAP age-depth model (*Simonsen et al., 2019*)

L101 Were samples transported frozen?

The samples were melted for sampling and frozen immediately after sampling. They were then kept frozen and away from light sources until they were analysed.

L102 Replace "of" with "at"

Done

L102 "Curtin University of Technology" is now "Curtin University". Please replace here and throughout the manuscript. See https://www.curtin.edu.au/

Done

L105 Change "CUT" to "CU" here and throughout.

Done

L105 Please add reference to the Curtin University ICP-MS method or if these are the first measurements for iodine reported from that lab then please add methods and a table in the supplement with the operating parameters (e.g. torch, spray chamber, nebuliser).

The details on the Curtin University ICP-MS method is fully described in Maffezzoli et al., 2018, we have added that reference in the new version of the manuscript.

L106 You define the Italian ICP-MS as "IDPA-CNR" please make sure you are consistent with your terminology throughout the manuscript.

Done

L107 Reference is repeated twice. Please delete one.

Done

L107 Add resolution for sodium measurements.

This information has been added in the section 2.2 of the revised manuscript (Pag 4, lines 122-125).

L109 Spell out ultrapure water "UPW" acronym.

Done

L109 What are these instrumental errors? Are they related to reproducibility or accuracy? Please add accuracy and precision for iodine and sodium measurements. What certified reference material did you use?

The error bars on each data point reflect the standard deviation (1 sigma) of the set of 5 mass scans performed by the ICP-MS for each sample detection. Therefore, they reflect the single-measurement precision. As Standard Reference Material we used one of the standards (NIST traceable commercial standards: High-Purity Standards, (Charleston, USA)) used during the calibrations as Quality Controlled Standards (QCs, as described in Maffezzoli et al. (*2018*, Appendix A1), continuously monitored during the analyses, for a total of n=82 detections. The spread (relative standard deviation) of the n=82 QCs concentrations (precision) was: 16% (iodine); 4% (sodium). The accuracy (as the ratio between the measured QCs concentration and the nominal QCs concentration) was on average 95±29% (iodine); 98±7% (sodium), See plots below.

[Figure]

[Figure]

**Suplementary figure A**: Iodine and sodium accuracy in the measured samples.

L110-111 Use consistent terminology for Italian and Australian ICP-MS.

Done

L113 Yes there is a strong correlation between the two labs but it is non-linear at low concentrations. Include a note on the large area at low iodine concentrations.

We agree with reviewer on the non-linear relation at low concentrations and we have added a note in the text regarding this (Pag 4, lines 119-121). Nevertheless, the HTM iodine concentrations would still be significantly higher than the Neoglacial and the late Holocene iodine values, even accounting for the concentration non-linearity.

L116 Add "our" between" model and...sampling".

Done

L116 Add "mass" between "depositional...fluxes".

Done

L118 Please add reference for accumulation rates.

Done

L119 Please add the different climatic phases in the introduction.

Done

L121 Calcium is not mentioned previously. Either include Ca methods or remove entirely.

We have added Ca methods in the new version of the manuscript.

L124MBL spell out acronym.

Done

L128 First mention of photolytic processes. Need to include iodine snow photochemistry in the introduction.

In former line 128 (within the Atmospheric Chemistry modelling section) we were referring to the photochemistry in the gas phase, not to the photochemistry in the snowpack. In this work, we are using THAMO to estimate the amounts of inorganic and organic iodine emissions that would reproduce the tropospheric iodine levels recorded in the ice core. Therefore, we are referring to the photolysis of the different species emitted to the troposphere, and more specifically to the photolysis of the different reactive iodine precursors (inorganic: HOI, $I_2$ and organic: $CH_3I$, $CH_2I_2$, $CH_2IBr$ and $CH_2ICl$). Nevertheless, we have included a note in the supplementary information regarding the post-depositional processes.

L132 HTM spell out acronym.

We have spelled out the acronym in Pag 1 (line 23) in the new version of the manuscript.

L132Delete "rs" in ""ky-1".

Done

L135 Replace "results" with "ice core and modelling results" and add a fuller explanation of the modelling results here. Could you add a figure or table summarising the results?

We have modified this heading as "Results and Discussion" in the new version of the manuscript. We have added Table S1 into the main text and we have added a new figure in the supplementary material summarizing the THAMO modelling main results

L137-138 Symbolµ.

Corrected

L139 What causes re-mobilisation?

There are several wide range of factors that may contribute to iodine remobilization from surface snow and ice (e.g. photo-activation of reactive iodine, ice/firn characteristics, effect of snowfall rates, changes in meteorological conditions, etc) summarized in the literature (*Frieb et al., 2000; Spolaor, et al 2013; Galvez et al., 2016; Kim et al., 2016; Legrand et al., 2018, Cuevas et al., 2018; Spolaor et al., 2019*). The different mechanisms controlling iodine re-mobilization have been fully explained in a full section (SI) in the revised version of the manuscript.

L140 Please add reference of iodine concentration and accumulation rate studies. You haven't quantified how the accumulation rate impacts iodine loss at the Reland site so please include some evidence to justify that your assumption that post-depositional processes are negligible.

*We have added key references to the different accumulation rate studies in Greenland (e.g. Maselli et al., 2017, Rhodes et al., 2017). As discussed above, we have added a new section in the revised version of the manuscript regarding post-depositional effects on ReCAP site.*

L154 Symbolµ.

*Corrected*

L147 What are the biomarkers?

*Biomarkers are summarized in Table S2 and Fig. 1 and explained below in the text.*

L166 "The increase in nutrient supply from terrestrial sediment delivery".

*This has been corrected in the new version of the manuscript.*

L172 Replace "The"with "An".

*Done*

L174 Can you provide a reference "biological iodine explosion".

*"Biological Iodine Explosion" is a term suggested in this study for the first time.*

L184 ssa acronym.

*ssa acronym has been previously spelled out in line 57 of the former version of the manuscript.*

L185 Please mention calcium dust proxy and add reference.

*We have mention calcium as a dust proxy in Greenland and provided key references (i.e. (Schüpbach et al., 2018))*

L185-186 Is this result from your model?

*Yes, we have specified it in the text now*

L193 Replace "on" with "of".

*We have not modified it since we consider " on" is more appropriate in this case*

L186 Period.

*Ok, we have rephrased it accordingly*

L194 Replace"value" with "level".

*Done*

L195 Add location of Kara and Laptev Seas to Fig. 1.

*Their location cannot be shown in Fig. 1 since they are located in the Russian Arctic. We have explained its location in the text in the new version of the manuscript (Pag 7, lines 221).*

L200 What measure of sea ice referring to? Sea ice extent or concentration or thickness? Here and throughout manuscript please specify.

Sea ice extent and thickness, we have clarified this throughout the manuscript

L204 How much thicker does the sea ice need to be? Can you give estimates of thickness and light penetration depth?

In Antarctica, the light transmitted through sea-ice (with an average depth of ~50 cm) has significant transmission down to ~1m (*King et al., 2005*). This contrast with the sea-ice thickness in the Arctic (~3 m), where iodine emissions are not significant (*Saiz-Lopez et al., 2015*). Therefore, a sea-ice thickness of ~3 m would lead to negligible emissions of iodine from sea ice. We have now mention this in the manuscript (Pag 7, lines 229).

L204 Add sea ice as a source of iodine in the introduction.

Sea-ice as a source of iodine is now mentioned in the introduction in the new version of the manuscript explaining the biogenic production of HOI and I- from algae underneath the sea ice and its subsequent diffusion through brine channels to the atmosphere (Pag 2, lines 46-48).

L211 Should the fluxes and concentrations be reported as 2 significant figures?

Iodine concentration and fluxes curves are shown in Fig. 2 and in Figs. S2 and S5 for further details. Both fluxes and concentrations are shown in the same figures to allow intercomparison between them.

L222 Can you quantify the frequency? L223 What is the second part?

We have rephrased the sentence as follows in order to make it more readable

"*The iodine time series in this part of the record reveals a higher variability mainly due to a reduced ice compression towards the surface and thus an increased temporal resolution of the samples*"

L247 What are the associated radiative impacts?

In this line we are referring to the impacts on the radiative balance of the atmosphere and therefore on climate. These impacts are on tropospheric ozone (Hossaini et al., 2015; Saiz-Lopez et al., 2012; Sherwen et al., 2017) and particle formation (Allan et al.,2015; Roscoe et al., 2015, Sipilä et al., 2016), as already commented at the end of the first paragraph of the Introduction section. We have now included these impacts in the discussion and provided the related references (Pag 9, lines 275-278).

Figure 1 Mention the references in the caption to help the reader.

Done

Figure S1 What does the 1:1 line indicate? Be consistent with the naming of the Australian and Italian iodine measurements.

The 1:1 line indicates the x=y relation (it is not a fit). We have modified the naming according to reviewer suggestion

**References**

Allan, J. D., Williams, P. I., Najera, J., Whitehead, J. D., Flynn, M. J., Taylor, J. W., Liu, D., Darbyshire, E., Carpenter, L. J., Chance, R., Andrews, S. J., Hackenberg, S. C., and McFiggans, G.: Iodine observed in new particle formation events in the Arctic atmosphere during ACCACIA, Atmos. Chem. Phys., 15, 5599-5609, 2015

Cuevas, C. A., Maffezzoli, N., Corella, J. P., Spolaor, A., Vallelonga, P.,Kjær, H. A., Simonsen, M., Winstrup, M., Vinther, B., and Horvat, C.: Rapid increase in atmospheric iodine levels in the North Atlantic since the mid-20th century, Nature communications, 9, 1452, 2018.

Frieß, U., Deutschmann, T., Gilfedder, B., Weller,R., and Platt, U.: Iodine monoxide in the Antarctic snowpack, Atmospheric Chemistry and Physics, 10, 2439-2456, 2010.

Gálvez, Ó., Baeza-Romero, M. T., Sanz, M., and Saiz-Lopez, A.: Photolysis of frozen iodate salts as a source of active iodine in the polar environment, 2016.

Hossaini, R., Chipperfield, M., Montzka, S., Rap, A., Dhomse, S.and Feng, W., Efficiency of short-lived halogens at influencing climate through depletion of stratospheric ozone, Nature Geoscience, 8(3), 186, 2015.

Kim, K., Yabushita, A., Okumura, M., Saiz-Lopez, A., Cuevas, C. A., Blaszczak-Boxe, C. S., Min, D., W. Yoon, H.-I. and Choi, W., Production of molecular iodine and tri-iodide in the frozen solution of iodide: implication for polar atmosphere, Environmental science & technology, 50(3), 1280-1287, 2016.

King, M. D., France, J. L., Fisher, F. N., & Beine, H. J.. Measurement and modelling of UV radiation penetration and photolysis rates of nitrate and hydrogen peroxide in Antarctic sea ice: An estimate of the production rate of hydroxyl radicals in first-year sea ice. Journal of photochemistry and photobiology A: Chemistry, 176(1-3), 39-49. 2005.

Laskar, J., P. Robutel, F. Joutel, M. Gastineau, A. Correia, B. Levrard, A long-term numerical solution for the insolation quantities of the Earth. Astronomy & Astrophysics 428, 261-285, 2004.

Legrand, M., McConnell, J. R., Preunkert, S., Arienzo, M.,Chellman, N., Gleason, K., Sherwen, T., Evans, M. J., and Carpenter, L. J.: Alpine ice evidence of a three-fold increase in atmospheric iodine deposition since 1950 in Europe due to increasing oceanic emissions, Proceedings of the National Academy of Sciences, 115, 12136-12141, 2018.

Maffezzoli, N., Vallelonga, P., Edwards, R., Saiz-Lopez, A., Turetta, C., Kjær, H. A., Barbante, C., Vinther, B. and Spolaor, A., 120,000 year record of sea ice in the North Atlantic, Climate of the Past Discussions, https://doi.org/10.5194/cp-2018-80, 2018.

Maselli, O. J., Chellman, N. J., Grieman, M., Layman, L., McConnell, J., Pasteris, D., Rhodes, R. H., Saltzman, E. and Sigl, M. Sea ice and pollution-modulated changes in Greenland ice core methane sulfonate and bromine. Climate of the Past, 13 (1). pp. 39-59. 2017

Rhodes, R. H., Yang, X., Wolff, E. W., McConnell, J. R., & Frey, M. M. Sea ice as a source of sea salt aerosol to Greenland ice cores: a model-based study. Atmospheric Chemistry and Physics, 17(15), 9417-9433. 2017

Roscoe, H. K., Jones, A. E., Brough, N., Weller, R., Saiz-Lopez, A., Mahajan, A. S., Schoenhardt, A., Burrows, J. P., and Fleming, Z. L., Particles and iodine compounds in coastal Antarctica, Journal of Geophysical Research: Atmospheres, 120(14), 7144-7156, 2015

Saiz-Lopez, A., Plane, J. M., Baker, A. R., Carpenter, L. J., von Glasow, R., Gómez Martín, J. C., McFiggans, G. and Saunders, R. W., Atmospheric chemistry of iodine, Chemical reviews, 112(3), 1773-1804, 2012.

Saiz-Lopez, A., Blaszczak-Boxe, C.S. and Carpenter, L., A mechanism for biologically induced iodine emissions from sea ice, Atmospheric Chemistry and Physics, 15(17), 9731-9746, 2015.

Schüpbach, S., Fischer, H., Bigler, M., Erhardt, T., Gfeller, G., Leuenberger, D.et al. Greenland records of aerosol source and atmospheric lifetime changes from the Eemian to the Holocene. Nature communications, 9(1), 1476. 2018

Sherwen, T., Evans, M. J., Carpenter, L. J., Schmidt, J. A., and Mickley, L. J., Halogen chemistry reduces tropospheric O3 radiative forcing, Atmos. Chem. Phys., 17, 1557-1569, 2017.

Simpson, W. R., Alvarez-Aviles, L., Douglas, T. A., Sturm, M., and Domine, F. Halogens in the coastal snow pack near Barrow, Alaska: Evidence for active bromine air-snow chemistry during-springtime. Geophysical research letters, 32(4), 2005.

Sipilä, M., Sarnela, N., Jokinen, T., Henschel, H., Junninen, H., Kontkanen, J., Richters, S., Kangasluoma, J., Franchin, A. and Peräkylä, O., Molecular-scale evidence of aerosol particle formation via sequential addition of HIO3, Nature, 537(7621), 532, 2016.

Simonsen, M.F., Baccolo, G., Blunier, T., Borunda, A., Delmonte, B., Frei, R., Goldstein, S., Grinsted, A., Kjær, H.A., Sowers, T., Svensson, A., Vinther, B., Vladimirova, D., Winckler, G., Winstrup, M. and Vallelonga, P. East Greenland ice core dust record reveals timing of Greenland ice sheet advance and retreat. Nature Communications, 10(1), 4494, 2019.

Spolaor, A., Vallelonga, P., Plane, J., Kehrwald,N., Gabrieli, J., Varin, C., Turetta, C., Cozzi, G., Kumar, R., and Boutron, C.: Halogenspecies record Antarctic sea ice extent over glacial–interglacial periods, Atmospheric Chemistry and Physics, 13, 6623-6635, 2013.

*Spolaor, A., P. Vallelonga, J. Gabrieli, T. Martma, M. P. Björkman, E. Isaksson, G. Cozzi, C. Turetta, H. A. Kjær, M. A. J. Curran, A. D. Moy, A. Schönhardt, A. M. Blechschmidt, J. P. Burrows, J. M. C. Plane and C. Barbante. Seasonality of halogen deposition in polar snow and ice. Atmos. Chem. Phys. 14: 9613-9622. 2014.*

*Spolaor, A., Barbaro, E., Cappelletti, D., Turetta, C., Mazzola, M., Giardi, F., Björkman, M.P., Lucchetta, F., Dallo, F., Pfaffhuber, K.A., Angot, H., Dommergue, A., Maturilli, M., Saiz-Lopez, A., Barbante, C. and Cairns, W.R.L. Diurnal cycle of iodine and mercury concentrations in Svalbard surface snow. Atmos. Chem. Phys. Discuss. 2019, 1-25. 2019*

**Referee #2**

In this study, the first ice core iodine record spanning the Holocene is presented. Iodine levels were high during the early Holocene and industrialization relative to the late Holocene. Chemical transport model results of inorganic iodine sources and transport are compared to ice core iodine levels. This comparison suggests that marine biogenic sources of iodine were higher during the early Holocene relative to the late Holocene. This increase in biogenic sources is reasonable in the early Holocene given that other reconstructions show that sea surface temperatures in the North Atlantic were warmer, salinity was higher, sea ice extent was lower, and primary productivity of subpolar planktonic species was higher. This study is well-written and well-organized. It should be published because it is the first record iodine covering the early Holocene and provides strong links to climatic factors that could explain the unexpectedly high levels of early Holocene iodine. Hopefully, it will inspire future study of iodine species in ice cores.

We appreciate the positive feedback from referee #2. Please, find below the responses to all the comments made by the reviewer

Major comments

Section 1, lines 36-37: This sentence is confusing. What exactly is meant by "has a global contribution of up to 27% of the total rate of ozone loss?"

We are referring to reactive iodine as an important ozone-depleting family in the troposphere. Specifically, we are saying that reactive iodine is responsible of up to 27% of the ozone loss in both the marine boundary layer and the upper troposphere. We have rephrased that statement to clarify the meaning (Pag 2, lines 36-38).

Section 1, lines 58-68: Two ice cores are reported as having the only iodine records, but further in the paragraph, a third ice core record is discussed. Please clarify.

We have modified the sentence to clarify the iodine reconstruction in polar regions available up to date (Pag 2, lines 59-61).

Section 2.1, line 91-96. Please discuss the depth-age scale in more detail because (it is not yet published?). Perhaps include a depth-age scale in the supplement. Which volcanic markers were used and how well-spaced are they throughout the record? What are the uncertainties?

The ReCAP ice core record covers the last 120 kyr BP and has been published in Simonsen et al. (*2019*). The RECAP timescale down to 458.3 m (4048.1 y b2k) was produced using the StratiCounter automated layer counting software (https://github.com/maiwinstrup/StratiCounter) (*Winstrup, 2016*). The software was constrained to fit 28 volcanic marker horizons dated in other Greenland ice cores (DYE-3, GRIP, GRIP and NEEMS1) formalised in the Greenland Ice Core Chronology 2005 (*Rasmussen et al., 2013*). Below 458.3 m, the timescale

was derived automatically using a shape-preserving piecewise cubic interpolation (*Vinther, et al., 2008*) between 15 chronological tie points (volcanic markers or climate transition matches) used to constrain the timescale back to 11703 y b2k. As the RECAP timescale is synchronised to GICC05, it inherits the uncertainty budget associated with that timescale and presented in the aforementioned references.

Section 2.2, lines 99-100. Please include the sampling resolution at various depths. Clearly, several more samples were analysed in more recent years which makes sense as the fractions were collected via the CFA system. The easiest way to do this may be to include a mean sampling resolution for the time periods discussed in each section (results).

The iodine and sodium measurement temporal resolution ranged from sub-annual in the upper metres during the Great Acceleration (average resolution of 1.31 samples/yr), subdecadal resolution during the Late Holocene (average resolution of 0.23 samples/yr), and sub-centennial resolution during the Neoglacial and the HTM (average resolution of 0.035 and 0.011 samples/yr respectively). This information has been added in the revised manuscript (Pag 4, lines 122-125).

Section 3: Please use the same units when possible. It is difficult for the reader to understand readily the comparisons between different periods of time and the model versus the observations when the units are changed.

We have now changed the unit of the modelling results throughout the whole text from nmol m$^{-2}$ d$^{-1}$ to µg m$^{-2}$ y$^{-1}$ according to the observations units.

Section 3: It would be useful to discuss the relationship between concentration and flux here. Did the flux calculation confirm that remobilization processes were not consequential? Are there any important differences when flux is used?

Accumulation rates in ReCAP are much higher than other Greenland sites summarized in Maselli et al (*2017*) and Rhodes et al (*2017*) we may suggest that wet deposition dominates in ReCAP site. Nevertheless, this relation is not straight forward and we prefer to add both concentration and fluxes. According to Legrand et al. (*2018*) iodine concentration in Greenland ice are also dependent on the annual snowfall rates. In ReCAP ice core, reconstructed accumulation rates shown in Figure S2 shows reasonably stable levels for the last 8000 years, suggesting relatively constant snowfall and surface conditions. Variability in the accumulation occurring prior to 8000 years ago may be due to glaciological flow effects already identified in the previous Renland ice core drilled in 1988 (*Hansson et al., 1994*) just 2 km from the site of the 2015 RECAP ice core (this work).The absence of any significant correlation between the accumulation rates and the concentration of impurities such as iodine and ssa also indicate that the Holocene iodine variability in ReCAP ice core is neither related to the effects of changing snow deposition rates. Furthermore, both iodine concentration and fluxes shown similar trends. This agreement between concentration and fluxes reinforces our explanation of external forcings (i.e. marine

bioproductivity and sea ice dynamics, driving the iodine levels variability throughout the Holocene.

Line 141: Start a new paragraph here? How does the second part of this paragraph relate to the first part?

Done, we have modified this section by adding a new section in the SI regarding re-mobilization processes in snow and ice

3.1, lines 182-183 the authors state that there is no correlation. Is it possible to also include a time series of calcium and sodium measurements in the supplement?

Correlation coefficients of sodium and calcium are shown in Table S2 and in section. Sodium profiles are shown in Maffezzoli et al (2018) while calcium dataset is available upon request. We have stated this in the revised manuscript (Pag 9, line 296).

Lines 191: Please use the same units throughout the manuscript when possible.

Done

Lines 192-193: How would the ssa contribution change over time? Is the lack of a correlation during the early Holocene purely due to the biogenic contribution overwhelming the other signals? Are they still there?

Sea salt aerosol (ssa) inputs to RECAP are a combination of sea salt directly lofted from open ocean (wave breaking and storms) as well as sea salt incorporated into blowing snow over sea ice. When iodine is primarily emitted through organic processes there should be less correlation between ssa and iodine. When iodine is primarily emitted through inorganic processes, there should be a greater correlation. The reviewer's description is therefore accurate, that the lack of correlation during the early Holocene supports the interpretation of a dominance of organic iodine emission sources. As described in Table 1, the THAMO model results indicate organic iodine emissions were lower during the Neoglacial, but were higher before and after the Neoglacial and continue to the present. Ssa showed an abrupt increased at the beginning of the Holocene and fluctuating values throughout the Holocene. This variability is described in Maffezzoli et al. (2018). During the HTM the biogenic contribution would be much higher than the iodine related to atmospheric ssa deposition. This is well reflected in the I/Na ratio that have been added to new Fig. S4 in the revised version of the manuscript. However, ssa, as well as dust, could also be minor contributors of the iodine levels during the HTM. This has been reflected in the revised manuscript (Pag 5, lines 157-158; Pag 7, lines 208-211).

Lines 205-208: Please include the ice core values here and clarify how this conclusion was made, given the values provided.

We have added the mean iodine depositional fluxes from the ice core during this period. The modelling results during this period show that inorganic iodine emissions were significantly higher than the organic iodine emissions (Table 1) which is in agreement with the decreased in bioproductivity shown in Fig. 2 and described in the text. We have now provided the values for organic iodine emissions in the new version of Table 1 in order to make this clearer

Lines 221-223: Replace "a higher frequency" with "higher frequency variability." I agree that this is pretty clearly due to increased sampling resolution. If anything, averaging or smoothing could be used to compare the late Holocene to the early Holocene. The late Holocene levels may only appear to show higher frequency variability because there are more samples.

We agree with reviewer. Thus, we have rephrased the sentence as follows:

"*The iodine time series in this part of the record reveals a higher frequency variability mainly due to a reduced ice compression towards the surface and thus an increased temporal resolution of the samples*"

The focus of this study is iodine and should remain iodine. Do other halogens, like bromine, that were measured in the ReCAP ice core (Maffezzoli et al., 2018, in re-view), also show the same high levels during the early Holocene? Are the processes governing the sources, transport, and deposition of other halogens so different that they should not be included in the supplement?

Neither Br nor Cl show that abrupt difference between Early Holocene and the rest of the Holocene. Bromine enrichments found in ReCAP ice core are mostly influenced by the autocatalytic processes in Multi-Year Vs First-Year sea-ice that are fully described in Maffezzoli et al (*2018*). On the other hand, Cl is dependent on ssa variability and other complex recycling processes in the polar atmosphere that are beyond the scope of this paper. Therefore, in our opinion it would not make sense to add the other halogens in the supplementary material.

Consider adding a section about the modelling results. These results are used to explain the interpretation of the ice core record, but it would be useful to have an understanding of the results prior to the comparison.

We have elaborated on the modelling section. We have added more information about THAMO and described with more detail the outcome of the modelling results in the revised version of the manuscript (Pag 5, lines 131-152; Pag 7, lines 204-208; Pag 8, lines 234-238). On top of that we have added a new figure in the supplementary material (Figure S3) that display the modelled the atmospheric reactive iodine according to the different oceanic iodine emissions during the present day and the different Holocene periods discussed in the text.

[Figure]

**Figure 4. Diurnal variation of reactive iodine modelled by THAMO in different scenarios of [O₃] and VSLs e missions fluxes** (table 1), from present day (top left): [O₃]= 30 ppbv and present day VSL emission fluxes; Holocene Thermal Maximum (top right): [O₃] = 10 ppbv and 100% increase in present day VSL emission fluxes; Late Holocene (bottom left): [O₃] = 10ppbv and 50% decrease in present day VSL emission fluxes; Neoglacial (bottom right): [O₃] = 10 ppbv and 87% decrease in present day emission fluxes. I$_y$ comprises I$_2$+IO+HOI+HI+INO$_2$+INO$_3$

Figures and tables

fig. 1 add mapping software reference and color bar if going to include bathymetry

We have added the software reference (Ocean Data View, Schlitzer, R., 2015, http://odv.awi.de) and the colour bar in the new version of the figure.

[Figure]

**Fig. 1: Location of the ReCAP ice core (red) and other marine paleoceanographic archives in the Nordic Seas discussed in the text**.

fig.2 add indication of detection limits to iodine concentrations

We have added the information of the detection limits in the figure caption

Table S1: Please consider moving table S1 to the main text.

Done, Table S1 has been incorporated to the main text as Table 1. The table now includes the inorganic and organic oceanic emission in the same units of the ice core fluxes ($\mu g\ m^{-2}\ y^{-1}$).

Please add more to this caption. What are the sources (ice core versus model inputs/outputs) in each column?

We have now included more information in the caption of the table, defining the different fluxes included in the organic and inorganic iodine columns.

Minor Comments:

Abstract, line 22: change to: "in the record were found"

Done

line 30: "allows for the detailed"

Done

line 32: "key factor for understanding"

Done

line 42: change to "involve"

Done

 line 59: "Ice core"

Done

line 66: "allow for a"

Done

Rename section 2.1 "Study site and dating" or "Study site and age scale"

Done

Line 108: "with the stability of the instrumental"

Done

Line 126: remove "The model"

Done

Line 129: "all of the"

Done

Line 138: add comma before "respectively"

Done

Line 141: remove "thus" before "Holocene"

Done

Line 146: remove "occurring" before "before"

Done

Line 203: change to "algal"

Done

Line 205: change to "both of these processes"

Done

Line207, 220: "concentrations"

Done

Line 221: commas before and after "respectively"

Done

Line 241:comma before "resulting"

Done

Line 251: "to accelerate considerably"

Done

**References**

Hansson, M. E., The Renland ice core. A Northern Hemisphere record of aerosol composition over 120,000 years. Tellus B: Chemical and Physical Meteorology, 46(5), 390-418, 1994.

Legrand, M., McConnell, J. R., Preunkert, S., Arienzo, M.,Chellman, N., Gleason, K., Sherwen, T., Evans, M. J., and Carpenter, L. J.: Alpine ice evidence of a three-fold increase in atmospheric iodine deposition since 1950 in Europe due to increasing oceanic emissions, Proceedings of the National Academy of Sciences, 115, 12136-12141, 2018

Maffezzoli, N., Vallelonga, P., Edwards, R., Saiz-Lopez, A., Turetta, C., Kjær, H. A., Barbante, C., Vinther, B. and Spolaor, A., 120,000 year record of sea ice in the North Atlantic, Climate of the Past Discussions, https://doi.org/10.5194/cp-2018-80, 2018.

Maselli, Olivia J., Chellman, Nathan J., Grieman, Mackenzie, Layman, Lawrence, McConnell, Joseph R., Pasteris, Daniel, Rhodes, Rachael H., Saltzman, Eric and Sigl,Michael. Sea ice and pollution-modulated changes in Greenland ice core methane sulfonate and bromine. Climate of the Past, 13 (1). pp. 39-59. 2017

Rasmussen, S. O. et al. A first chronology for the North Greenland Eemian Ice Drilling (NEEM) ice core. Clim. Past 9, 2713-2730. 2013.

Rhodes, R. H., Yang, X., Wolff, E. W., McConnell, J. R., & Frey, M. M. Sea ice as a source of sea salt aerosol to Greenland ice cores: a model-based study. Atmospheric Chemistry and Physics, 17(15), 9417-9433. 2017

Schlitzer, R., 2015, Ocean Data View, http://odv.awi.de

Simonsen, M.F., Baccolo, G., Blunier, T., Borunda, A., Delmonte, B., Frei, R., Goldstein, S., Grinsted, A., Kjær, H.A., Sowers, T., Svensson, A., Vinther, B., Vladimirova, D., Winckler, G., Winstrup, M. and Vallelonga, P.. East Greenland ice core dust record reveals timing of Greenland ice sheet advance and retreat. Nature Communications, 2019, in press.

*Vinther, B. M. et al. Synchronizing ice cores from the Renland and Agassiz ice caps to the Greenland Ice Core Chronology,. J. Geophys. Res.* **113,** *D08115, doi:08110.01029/02007JD009143. 2008*

*Winstrup, M. A Hidden Markov Model Approach to Infer Timescales for High-Resolution Climate Archives.  (AAAI Press). 2016*

**Referee #3**

In the paper "Holocene atmospheric iodine evolution over the North Atlantic" by J. P.Corella and co-authors a new record of iodine concentration in the ReCAP ice core (Greenland) is presented. The record covers the full Holocene and an interesting discussion about iodine variations in the atmosphere across different climatic periods is shown. In particular the authors elaborate here an easy conceptual model which seems to be very effective in explaining iodine variability during the three time periods here considered: Holocene Thermal Maximum, Neoglacial period and Great Acceleration. The paper shows a very good laboratory intercomparison exercise, showing a general agreement in the measured iodine concentrations. This work is generally well presented and well written and will surely be a milestone in the use of halogens measured in ice cores to achieve information about Sea Ice in the North Atlantic.

We appreciate the positive feedback from referee #3. We have addressed below all the comments and suggestions made by reviewer.

My main concern about the paper is the lack of discussion about the possible post-depositional effects of iodine species in the snow. A detailed discussion about the preservation of iodine compounds in the snow and their stability in time due to a possible photochemical degradation should be added to the text. Before concluding that iodine is a real marker reflecting the past North Atlantic conditions, the authors should assess that the very low concentrations they are measuring are really telling us a story about past conditions, or, at least, that the record obtained is not significantly different from what was deposited in the Renland Ice Cap.

We agree with reviewer on the fact that photochemical processes on snow and ice lead to iodine recycling for the surface snow. Indeed, several laboratory and field experiments reported in the literature (e.g Frieb et al., 2000; Spolaor et al. 2014; Galvez et al., 2016; Kim et al., 2016; Spolaor et al., 2019) have shown that iodine concentrations from in polar regions may suffer from significant photo-chemical post-depositional processes that may affect the iodine concentration in ice and snow at daily to seasonal scales. Nevertheless, large variations in the net iodine depositional fluxes in coastal Greenland ice cores at centennial to millennial time scales are not expected. We have added a section in the new version of the manuscript (SI) where the post-depositional processes affecting iodine re-emission from ice and snow at different time scales are fully addressed

Moreover, the meteorological conditions were surely very different in the different time periods here shown, and the authors should clarify why the variability of iodine is mainly linked to the primary productivity and not to changes in transport efficiency through time.

It is difficult to reconstruct meteorological conditions in Greenland accurately during the Holocene. The best indicator we have of the conditions at the site is the reconstructed accumulation rate shown in Figure S2. This was reasonably stable for the last 8000 years, suggesting relatively constant snowfall and surface conditions. The absence of any significant correlation between the Holocene accumulation rate and the concentration of impurities such as iodine and ssa, is

the first indication that the observed pattern of those impurities is not dominated by snow deposition or meteorological effects.

The scale of variability exhibited in the RECAP Iodine record (both concentrations and fluxes) is further inconsistent with possible changes in transport efficiency. The iodine concentration changes by more than a factor of 4 within a few centuries during the Holocene, which is an enormous change. For comparison, best-available estimates suggest only a minor change in transport efficiency (much less than a factor of 4) over the glacial-interglacial transition for dust travelling to Greenland from deserts in central Asia (*Schüpbach et al., 2018*)

Other papers by the authors reported an halogen enrichment factor by subtracting the sea salt contribution. I'm not sure about which is the best way to proceed, but I would like to know why in this paper this elaboration was not considered. I suspect that an elaboration of the iodine values with sodium can infer to iodine a variability which mainly comes from Na+ concentrations. Once these points will be addressed I would recommend the publication of this paper.

As suggested by reviewer, we have added a new figure (Fig. S5) showing the iodine to sodium ratio. In this figure we show that the I/Na ratio show the same trends that the iodine concentration and fluxes throughout the Holocene reinforcing our interpretation that ssa would only represent a minor contributor explaining the iodine variability in ReCAP during the Holocene. We have added this new information in the revised manuscript (Pag 5, lines 157-158; Pag 7, lines 208-211).

[Figure]

**Fig. S5**: **ReCAP ice core iodine to sodium concentration ratio during the Holocene**

Minor issues.

Line 59: Change to "Ice core..."

Done

Line 109: explain what UPW means.

Done

Line 110: How were the d.l. calculated? Explain briefly how the two labs measured this figure of merit of the two methods.

Sodium and iodine detection limits were calculated as 3 times the standard deviation of n=80 blank values (DL=3σ), which are 1 ppb and 0.005 ppb for the Italian system and 1.1 ppb and 0.002 ppb for the Australian system respectively. This information has been added to the methods section in the new version of the manuscript (Pag 4, lines 111-117).

Line 116: Is there an age model for the RECAP ice core? Give more information on this point: how was the age scale built? Which absolute markers were used (i.e. volcanoes, tephra, etc)? Which is the uncertainty in the bottom part of the ice core?

The ReCAP ice core record covers the last 120 kyr BP and has been published in Simonsen et al., 2019 where all the details regarding the chronology are fully explained.

Addressing the specific questions made by reviewer:

-The RECAP timescale down to 458.3 m (4048.1 y b2k) was produced using the StratiCounter automated layer counting software (https://github.com/maiwinstrup/StratiCounter) (*Winstrup, 2016*).

-The software was constrained to fit 28 volcanic marker horizons dated in other Greenland ice cores (DYE-3, GRIP, GRIP and NEEMS1) formalised in the Greenland Ice Core Chronology 2005 (*Rasmussen et al., 2013*). Below 458.3 m, the timescale was derived automatically using a shape-preserving piecewise cubic interpolation (*Vinther, et al., 2008*) between 15 chronological tie points (volcanic markers or climate transition matches) used to constrain the timescale back to 11703 y b2k.

-As the RECAP timescale is synchronised to GICC05, it inherits the uncertainty budget associated with that timescale and presented in the aforementioned references.

Line 137-138: units are not readable...

This has been corrected in the new version of the manuscript

Line 141: remove the comma between m-2 and yr-1.

Done

Line 198: Two references by Telesinski et al. were published in 2014, please call them a) and b) in order to make clear which reference you are referring to in the text.

Done

Line 198: is c. for ca.?

YES, this error has been corrected.

Line 240: remove M.M. before Telesinski.

Done

Line 412: check the references: something wrong happened to the title of this paper.

The typos from the title reference title have been corrected now

Figure 2 caption: use "concentration and flux" or "concentrations and fluxes", not a mix

Done

Figure 2: during the HTM the authors' reconstruction seems to be very smoothed with respect to the marine records. Is it due to the poor resolution at this depth? Please discuss this point in the text.

Yes, this is due to the sampling lower resolution at this depth. This has been discussed in the new version of the manuscript (Pag 4, lines 112-125).

Figure 2 and text: is flux adding something new with respect to concentration? The two profiles are very similar and the authors should discuss this point in terms of wet/dry deposition mechanisms at the drilling site.

According to Wolff et al. (*2006*) if the accumulation rate is low, dry deposition dominates and fluxes are proportional to the air concentration, while if the accumulation rate is high, wet deposition dominates the dry deposition, the concentration is proportional to the air concentration. Therefore, since accumulation rates in ReCAP are much higher than other Greenland sites summarized in Maselli et al (*2017*) and Rhodes et al (*2017*) we may suggest that wet deposition dominates. Nevertheless, this relation is not straight forward and we prefer to add both concentration and fluxes. Both iodine concentration and fluxes shown similar trends. This agreement between concentration and fluxes reinforces our explanation of external forcings (i.e. marine bioproductivity and sea ice dynamics, driving the iodine levels variability throughout the Holocene.

**References**

Frieß, U., Deutschmann, T., Gilfedder, B., Weller,R., and Platt, U.: Iodine monoxide in the Antarctic snowpack, Atmospheric Chemistry and Physics, 10, 2439-2456, 2010

Gálvez, Ó., Baeza-Romero, M. T., Sanz, M., and Saiz-Lopez, A.: Photolysis of frozen iodate salts as a source of active iodine in the polar environment, 2016.

Kim, K., Yabushita, A., Okumura, M., Saiz-Lopez, A., Cuevas, C. A., Blaszczak-Boxe, C. S., Min, D., W. Yoon, H.-I. and Choi, W., Production of molecular iodine and tri-iodide in the frozen solution of iodide: implication for polar atmosphere, Environmental science & technology, 50(3), 1280-1287, 2016.

Maselli, Olivia J., Chellman, Nathan J., Grieman, Mackenzie, Layman, Lawrence, McConnell, Joseph R., Pasteris, Daniel, Rhodes, Rachael H., Saltzman, Eric and Sigl,Michael. Sea ice and pollution-modulated changes in Greenland ice core methane sulfonate and bromine. Climate of the Past, 13 (1). pp. 39-59. 2017

Rhodes, R. H., Yang, X., Wolff, E. W., McConnell, J. R., & Frey, M. M. Sea ice as a source of sea salt aerosol to Greenland ice cores: a model-based study. Atmospheric Chemistry and Physics, 17(15), 9417-9433. 2017

Simonsen, M.F., Baccolo, G., Blunier, T., Borunda, A., Delmonte, B., Frei, R., Goldstein, S., Grinsted, A., Kjær, H.A., Sowers, T., Svensson, A., Vinther, B., Vladimirova, D., Winckler, G., Spolaor, A., Vallelonga, P., Plane, J., Kehrwald,N., Gabrieli, J., Varin, C., Turetta, C., Cozzi, G., Kumar, R., and Boutron, C.: Halogenspecies record Antarctic sea ice extent over glacial–interglacial periods, Atmospheric Chemistry and Physics, 13, 6623-6635, 2013.

Spolaor, A., Barbaro, E., Cappelletti, D., Turetta, C., Mazzola, M., Giardi, F., Björkman, M.P., Lucchetta, F., Dallo, F., Pfaffhuber, K.A., Angot, H., Dommergue, A., Maturilli, M., Saiz-Lopez, A., Barbante, C. and Cairns, W.R.L. Diurnal cycle of iodine and mercury concentrations in Svalbard surface snow. Atmos. Chem. Phys. Discuss. 2019, 1-25. 2019

Schüpbach, S., Fischer, H., Bigler, M., Erhardt, T., Gfeller, G., Leuenberger, D.et al. Greenland records of aerosol source and atmospheric lifetime changes from the Eemian to the Holocene. Nature communications, 9(1), 1476. 2018

Winstrup, M. and Vallelonga, P.. East Greenland ice core dust record reveals timing of Greenland ice sheet advance and retreat. Nature Communications, 2019, in press.

Vinther, B. M. et al. Synchronizing ice cores from the Renland and Agassiz ice caps to the Greenland Ice Core Chronology,. J. Geophys. Res. **113,** D08115, doi:08110.01029/02007JD009143. 2008

Winstrup, M. A Hidden Markov Model Approach to Infer Timescales for High-Resolution Climate Archives. (AAAI Press). 2016.

Wolff, E. W., Fischer, H., Fundel, F., Ruth, U., Twarloh, B., Littot, G. C., ... et al. Southern Ocean sea-ice extent, productivity and iron flux over the past eight glacial cycles. Nature, 440(7083), 491. 2006